# CAN LLMS UNDERSTAND TIME SERIES ANOMALIES?

**Zihao Zhou**
Dept of Computer Science and Engineering
University of California, San Diego
La Jolla, CA 92093, USA
`ziz244@ucsd.edu`

**Rose Yu**
Dept of Computer Science and Engineering
University of California, San Diego
La Jolla, CA 92093, USA
`roseyu@ucsd.edu`

## ABSTRACT

Large Language Models (LLMs) have gained popularity in time series forecasting, but their potential for anomaly detection remains largely unexplored. Our study investigates whether LLMs can understand and detect anomalies in time series data, focusing on zero-shot and few-shot scenarios. Inspired by conjectures about LLMs' behavior from time series forecasting research, we formulate key hypotheses about LLMs' capabilities in time series anomaly detection. We design and conduct principled experiments to test each of these hypotheses. Our investigation reveals several surprising findings about LLMs for time series: (1) LLMs understand time series better as *images* rather than as text, (2) LLMs do not demonstrate enhanced performance when prompted to engage in *explicit reasoning* about time series analysis. (3) Contrary to common beliefs, LLMs' understanding of time series *does not* stem from their repetition biases or arithmetic abilities. (4) LLMs' behaviors and performance in time series analysis *vary significantly across different models*. This study provides the first comprehensive analysis of contemporary LLM capabilities in time series anomaly detection. Our results suggest that while LLMs can understand trivial time series anomalies, we have no evidence that they can understand more subtle real-world anomalies. Many common conjectures based on their reasoning capabilities do not hold. All synthetic dataset generators, final prompts, and evaluation scripts have been made available in `https://github.com/rose-stl-lab/anomllm`.

## 1 INTRODUCTION

The remarkable progress in large language models (LLMs) has led to their application in various domains, including time series analysis. Recent studies have demonstrated LLMs' potential as zero-shot and few-shot learners in tasks such as forecasting and classification (Gruver et al., 2023; Liu et al., 2024c). However, the effectiveness of LLMs in time series analysis remains a subject of debate. While some researchers argue that LLMs can leverage their pretrained knowledge to understand time series patterns (Gruver et al., 2023), others suggest that simpler models can match or outperform LLMs (Tan et al., 2024).

This debate raises a fundamental question: Do LLMs truly understand time series? To address this question, we must look beyond models' predictive performance. Forecasting typically hinges on metrics like MSE, which can overlook deeper model understanding. A model that simply outputs a nearly constant line might still achieve a passable MSE but reveals little about its capacity to interpret dynamics. Shifting our focus to anomaly detection changes the game: it forces LLMs to pinpoint irregular behavior and thus tests whether they grasp the underlying patterns, not just how well they extrapolate an average.

In this paper, we present the first comprehensive investigation into LLMs' understanding of time series data through the lens of anomaly detection. We focus on state-of-the-art LLMs and multimodal LLMs (M-LLMs) across different anomaly types under controlled conditions. Our evaluation strategy incorporates multimodal inputs (textual and visual representations of time series), various prompting techniques, and structured output formats. The results are quantitatively assessed using the affinity F1 score. We provide empirical evidence to challenge existing conjectures and

claims about LLMs' time series understanding. Our findings reveal a more nuanced understanding of LLMs' capabilities and limitations in time series data, including:

- **Visual Advantage:** LLMs perform significantly better in anomaly detection when processing time series data as *images* rather than text tokens.
- **Limited Reasoning:** LLMs do not benefit from chain-of-thought reasoning when analyzing time series data. Their performance often decreases when prompted to explain their reasoning.
- **Alternative Processing Mechanisms:** Contrary to common beliefs, LLMs' understanding of time series does not stem from their repetition biases or arithmetic abilities, challenging prevailing assumptions about how these models process numerical data.
- **Model Heterogeneity:** Time series understanding and anomaly detection capabilities differ across various LLM architectures, highlighting the importance of model selection.

In summary, while LLMs can detect simple anomalies, their abilities to understand and reason about numerical time series are considerably limited. Our research calls for more caution in applying LLMs to time series and for controlled studies to examine LLMs' behavior for other data modalities.

## 2 RELATED WORK

**LLMs for Time Series Analysis.** LLMs have been applied to various time series analysis tasks, with recent literature establishing strong claims about their capabilities. Gruver et al. (2023) showed that LLMs such as GPT-3 and LLaMA-2 possess broad pattern extrapolation capabilities, enabling zero-shot time series forecasting by encoding time series as strings of numerical digits and achieving comparable performance to purpose-built models. Building on these claims, Liu et al. (2024c) proposed a Cross-Modal LLM Fine-Tuning framework, suggesting that LLMs can provide interpretable predictions while addressing the distribution discrepancy between textual and temporal input tokens in multivariate time series forecasting. Liu et al. (2024b) introduced Time-MMD, a multi-domain, multimodal time series dataset for LLM finetuning. For anomaly detection, Liu et al. (2024a) proposed AnomalyLLM, a knowledge distillation-based approach using GPT-2 as the teacher network. Zhang et al. (2024) provided a comprehensive survey of LLM applications in time series analysis. In the financial domain, Wimmer & Rekabsaz (2023) used vision language models, i.e., CLIP, but not M-LLMs to process visualizations of stock data for market change prediction.

However, these prevailing beliefs about LLMs' pattern extrapolation capabilities and interpretable predictions remain controversial. Zeng et al. (2023) argued that the permutation-invariant nature of self-attention mechanisms may lead to loss of critical temporal information. Tan et al. (2024) found that removing the LLM component or replacing it with a basic attention layer often improved performance in popular LLM-based forecasting methods, challenging the assumed benefits of LLMs' pattern recognition abilities. The interpretability of LLMs in time series analysis remains a challenge, as their reasoning capabilities are often opaque and difficult to interpret.

**Time Series Anomaly Detection.** Time series anomaly detection is a critical task in various domains, including finance, healthcare, and cybersecurity. Traditional methods rely on statistical techniques, while recent work has focused on developing deep learning-based approaches (Audibert et al., 2022; Chen et al., 2022; Tuli et al., 2022). Audibert et al. (2022) compared conventional, machine-learning-based, and deep neural network methods, finding that no family of methods consistently outperforms the others. Chen et al. (2022) proposed a deep variational graph convolutional recurrent network for multivariate time series anomaly detection. Tuli et al. (2022) introduced a transformer-based anomaly detection model with adversarial training and meta-learning.

However, recent studies have highlighted significant flaws in current time series anomaly detection benchmarks and evaluation practices (Wu & Keogh, 2021; Huet et al., 2022; Sarfraz et al., 2024). Wu & Keogh (2021) argued that popular benchmark datasets suffer from major flaws like triviality and mislabeling, potentially creating an illusion of progress in the field. Huet et al. (2022) pointed out that the classical F1 score fails to reflect approximate but non-overlapping detections, which are common in time series. Sarfraz et al. (2024) criticized the persistent use of flawed evaluation metrics and inconsistent benchmarking practices, suggesting that complex deep learning models may not offer significant improvements over simpler baselines in the semi-supervised setting.

One reason for using deep learning models in time series anomaly detection is their ability to bring prior knowledge on what constitutes normal behavior from pretraining on large-scale datasets. Large language models (LLMs) may offer a promising solution due to their strong zero-shot capabilities. However, there is currently a lack of systematic study of modern (M-)LLMs for time series anomaly detection, which we aim to address in this work.

**Multimodal LLMs (M-LLMs).** Multimodal LLMs combine text with other data modalities and have been explored in various domains, including image captioning, video understanding, and multimodal translation (Lu et al., 2019; Li et al., 2019; Sun et al., 2019; Huang et al., 2019). Recent advancements have led to more sophisticated M-LLMs, such as Qwen-VL and Phi-3-Vision, demonstrating superior performance in visual-centric tasks and compact deployment capabilities (Bai et al., 2023; Abdin et al., 2024). In the context of time series analysis, M-LLMs have been used to model multimodal data, such as time series and textual information, showing promising results in forecasting and anomaly detection (Liu et al., 2021). However, there is a notable absence of research applying M-LLMs to time series data presented as visual inputs, even though humans often detect time series anomalies through visual inspection. This gap is particularly significant given that time series data can be represented in different modalities (e.g., numerical, textual, or visual) without losing substantial new information. Consequently, time series analysis presents a unique opportunity to evaluate M-LLMs' ability to understand and process the same underlying data across different representational formats.

## 3 TIME SERIES ANOMALY DETECTION: DEFINITION AND CATEGORIZATION

We begin by defining time series anomaly detection and categorizing different types of anomalies.

### 3.1 ANOMALY DEFINITION

We consider time series $X := \{x_1, x_2, \ldots, x_T\}$ collected at regular intervals, where $x_t$ is the feature scalar or vector at time $t$, and $T$ is the total number of time points. Anomalies are data points that deviate significantly from the expected pattern of the time series. The expected pattern of a time series refers to the governing function or conditional probability that the data is expected to follow, depending on whether the system is deterministic or stochastic.

*Generating function.* Assume the time series generation is deterministic. A data point $x_t$ is considered an anomaly if it deviates much from the value predicted by the generating function, i.e.,

$$|x_t - G(x_t|x_{t-1}, x_{t-2}, \ldots, x_{t-n})| > \delta \tag{1}$$

*Conditional probability.* Assume the time series generation is governed by a history-dependent stochastic process. A data point $x_t$ is considered an anomaly if its conditional probability is below a certain threshold $\epsilon$, i.e.,

$$P(x_t|x_{t-1}, x_{t-2}, \ldots, x_{t-n}) < \epsilon \tag{2}$$

An anomaly detection algorithm typically takes a time series as input and outputs either *binary labels* $Y := \{y_1, y_2, \ldots, y_T\}$ or *anomaly scores* $\{s_1, s_2, \ldots, s_T\}$. In the case of binary labels, $y_t = 1$ indicates an anomaly at time $t$, and $y_t = 0$ indicates normal behavior. The number of anomalies should be much smaller than the number of normal data points, i.e., $\sum_{t=1}^{T} y_t \ll T$.

In the case of anomaly scores, $s_t$ represents the degree of anomaly at time $t$, with higher scores indicating a higher likelihood of an anomaly. This likelihood can be connected to the conditional probability definition, where a higher score is correlated to a lower conditional probability $P(x_t|x_{t-1}, x_{t-2}, \ldots, x_{t-n})$. A threshold $\theta$ can be applied to the scores to convert them into binary labels, where $y_t = 1$ if $s_t > \theta$ and $y_t = 0$ otherwise.

Time series anomalies can be analyzed at two levels: (1) within individual sequences, where specific points or intervals deviate from the normal pattern, and (2) across different sequences, where entire sequences are considered anomalous. This paper focuses on the first level, specifically detecting anomalous intervals within individual sequences.

**Anomalous Intervals.** We define anomalies as continuous intervals of time points that deviate from the expected pattern. Let $R$ be a set of anomalous time intervals: $R = \{[t^1_{start}, t^1_{end}], [t^2_{start}, t^2_{end}], \ldots, [t^k_{start}, t^k_{end}]\}$ where $[t^i_{start}, t^i_{end}]$ represents the $i$-th anomalous interval, with $t^i_{start}$ and $t^i_{end}$ being its start and end times. When $t^i_{start} = t^i_{end}$, the anomaly is a single point anomaly. For a time series $X = \{x_1, x_2, \ldots, x_T\}$, we assign binary labels:

$$y_t = \begin{cases} 1 & \text{if } t \in [t^i_{start}, t^i_{end}] \text{ for any } i \in \{1, \ldots, k\} \\ 0 & \text{otherwise} \end{cases}$$

**Zero-Shot and Few-Shot Anomaly Detection.** Few-shot anomaly detection involves providing the model $f$ with a small set of labeled examples. Given $n$ labeled time series $\{(X_1, Y_1), (X_2, Y_2), \ldots, (X_n, Y_n)\}$, where $Y_i$ is a series of anomaly labels for each time step in $X_i$, and a new unlabeled time series $X_{new}$, the model $g$ predicts:

$$\{s_1, s_2, \ldots, s_T\} \text{ or } \{y_1, y_2, \ldots, y_T\} = g(X_{new}, \{(X_1, Y_1), (X_2, Y_2), \ldots, (X_n, Y_n)\}),$$

where $n$ is typically small (e.g., 1-5). In zero-shot anomaly detection, $n = 0$, i.e., the model $f$ is expected to identify anomalies without any labeled examples. These scenarios pose significant challenges for deep neural nets and some traditional models that require a lot of training examples.

## 3.2 ANOMALY PATTERN CLASSIFICATION

Patterns of time series anomalies can be categorized into two main types based on their nature: out-of-range anomalies which exceed normal value thresholds and contextual anomalies which exhibit abnormal behavior only within specific contexts (Lai et al., 2023). The contextual anomalies can be further divided into frequency anomalies, trend anomalies, and contextual point anomalies. Each type presents unique characteristics and challenges for detection. By examining how LLMs recognize these diverse anomaly types, we can verify whether our hypotheses about LLMs' understanding of time series data hold consistently across different pattern variations.

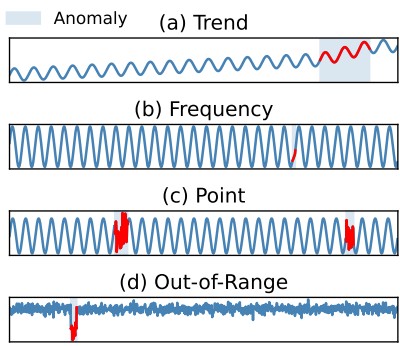

Figure 1: Example time series with different anomaly types, with anomalous regions highlighted in red.

### 3.2.1 OUT-OF-RANGE ANOMALIES

Out-of-range anomalies are data points that lie far outside the normal range of values in a time series. These anomalies can be detected even when the time series order is shuffled, as shown in Figure 1(d). If a model can detect out-of-range anomalies but fails to detect contextual anomalies, this suggests it is not using the positional information in the time series (Tan et al., 2024).

### 3.2.2 CONTEXTUAL ANOMALIES

Contextual anomalies are data points or consecutive subsequences that deviate from the expected pattern of the time series. These anomalies are only detectable when the order of the time series is preserved. Contextual anomalies can be further divided into three subcategories:

**Trend Anomalies.** Trend anomalies manifest as a sudden acceleration (Figure 1(a)), deceleration, or reversal of the established trend. These anomalies are characterized by unexpected changes in the changing rate of the time series, detected through gradient analysis $g_t = (x_t - x_{t-1})/\Delta t$. To reduce noise sensitivity, smoothed gradients $g_t^{smooth} = (MA_t - MA_{t-1})/\Delta t$ are often used, where $MA_t$ is a moving average over a window of points.

**Frequency Anomalies.** Frequency anomalies occur when the periodic components of a time series deviate from the expected pattern. These anomalies are usually identified by analyzing the frequency domain of the time series, typically using techniques like Fourier transforms. A frequency anomaly is detected when there's a significant shift in the dominant frequencies, see Figure 1(b).

**Contextual Point Anomalies.** Contextual point anomalies occur when individual data points deviate from the expected pattern of the time series, even while remaining within the overall regular

range of values. As shown in Figure 1(c), these points are not extreme outliers but don't fit the local context of the time series. They may violate the smooth continuity of the data, contradict short-term trends, or disrupt established patterns without necessarily exceeding global thresholds.

### 3.3 TIME SERIES FORECASTING VS. ANOMALY DETECTION

As most of the literature on LLM for time series focuses on forecasting, we use the conjectures from these works as a starting point to understand LLMs' behavior in anomaly detection. The tasks of time series forecasting and anomaly detection share many similarities. By definition, deterministic forecasting of future time steps is about learning the generating function (see Equation 1). Probabilistic forecasting is about learning the conditional probability function (see Equation 2). Therefore, both time series forecasting and anomaly detection rely heavily on extrapolation. In forecasting tasks, models extrapolate past patterns to predict future values, extending the known series into unknown territory. Similarly, anomaly detection involves extrapolating the "normal" behavior of a time series to identify points that deviate significantly from this expected pattern.

LLMs are trained on a corpus of token sequences, $\{U_1, U_2, \ldots, U_N\}$, where $U_i = \{u_1, u_2, \ldots, u_{L_i}\}$ and $u_j$ is a token in the vocabulary $\mathcal{V}$. The model learns to autoregressively predict the next token in the sequence given the previous tokens, i.e., $P(u_{j+1}|u_1, u_2, \ldots, u_j)$. The motivation for applying LLMs to time series forecasting is often their zero-shot extrapolation capabilities (Brown et al., 2020). The autoregressive generation of tokens and that of time series steps (in Equation 2) are similar, and the LLMs act as an Occam's razor to find the simplest form of $G$ (in Equation 1) (Gruver et al., 2023). This connection suggests that hypotheses made in LLMs for forecasting may also apply to anomaly detection. Many such hypotheses are proposed as possible explanations for the model's behavior without validation from controlled studies, which motivates our investigation.

## 4 UNDERSTANDING LLM'S UNDERSTANDING OF TIME SERIES

To demystify LLMs' anomaly detection capabilities, we take a principled approach by formulating several scientific hypotheses. Then we build an LLM time series anomaly evaluation framework to test each of the hypotheses.

### 4.1 HYPOTHESES

The following hypotheses represent a synthesis of existing literature (1-3), our own insights into LLM behavior (4-5), and prevailing assumptions in the field that warrant closer examination (6-7). The hypotheses cover two main aspects: LLMs' reasoning paths (1, 3, 4) and biases (2, 5, 6, 7).

> **Hypothesis 1 (Tan et al., 2024) on Chain-of-Thought (CoT) Reasoning**
>
> LLMs do not benefit from engaging in step-by-step reasoning about time series data.

The authors claim that existing LLM methods, including zero-shot ones, do very little to use innate reasoning. While they demonstrate that LLM methods perform similarly or worse than those without LLMs in time series tasks, none of their experiments assess the LLMs' reasoning capabilities.

To validate this hypothesis, we focus on the performance of one-shot and zero-shot CoT prompting, which explicitly elicits an LLM's reasoning abilities (Wei et al., 2022). We use terms from cognitive science to describe the LLMs' different behaviors with and without CoT: the reflexive mode (slow, deliberate, and logical) and the reflective mode (fast, intuitive, and emotional) (Lieberman, 2003). Therefore, the question becomes whether the LLMs benefit from the reflexive mode, when it thinks slowly about the time series. If the hypothesis is false, the LLMs should perform better when they are prompted to explain.

> **Hypothesis 2 (Gruver et al., 2023) on Repetition Bias**
>
> LLMs' repetition bias (Holtzman et al., 2020) corresponds precisely to their ability to identify and extrapolate periodic structure in the time series.

This hypothesis draws a parallel between LLMs' tendency to generate repetitive tokens and their potential ability to recognize periodic patterns in time series data. To validate the hypothesis, we design an experiment where the datasets contain both perfectly periodic and noisy periodic time series. The introduction of minor noise would disrupt the *exact* repetition of tokens in the input sequence, even if the underlying pattern remains numerically approximately periodic. If the hypothesis holds, we should observe a significant drop in performance, despite the numbers maintaining its fundamental periodic structure. See Appendix E for formal definitions of Hypothesis 2 and 3.

> **Hypothesis 3 (Gruver et al., 2023) on Arithmetic Reasoning**
>
> LLMs' ability to perform addition and multiplication (Yuan et al., 2023) maps onto extrapolating linear and exponential trends.

This hypothesis suggests a connection between LLMs' arithmetic capabilities and their ability to extrapolate simple mathematical sequences. It is argued that LLMs' proficiency in basic arithmetic operations, such as addition and multiplication, enables it to extend patterns like linear sequences (e.g., $x(t) = 2t$) by iteratively applying the addition operation (e.g., $+2$). To validate the hypothesis, we design an experiment where an LLM is specifically guided to *impair its arithmetic abilities* while preserving its other *linguistic and reasoning capabilities*. If the hypothesis holds, we should observe a corresponding decline in the model's ability to predict anomalies in the trend datasets. Otherwise, LLMs rely on alternative mechanisms for time series pattern recognition and extrapolation.

> **Hypothesis 4 (Dong et al., 2024) on Visual Reasoning**
>
> Time series anomalies can be more easily detected as visual input rather than text input.

Motivated by the fact that human analysts often rely on visual representations for anomaly detection in time series, we hypothesize that M-LLMs, whose training data includes human expert detection tasks, may prefer time series as images rather than raw numerical data. Similar assumption is also proposed in recent work by Dong et al. (2024), who demonstrated that explicitly prompting LLMs to "think visually" about time series improved their anomaly detection capabilities, even without actual visual input. This hypothesis can be readily tested by comparing the performance of multimodal LLMs on identical time series presented as both text and visualizations.

> **Hypothesis 5 on Visual Perception Bias**
>
> LLMs exhibit similar detection limitations to human perceptual biases, e.g., in acceleration perception when analyzing visual time series representations.

We hypothesize based on the growing interest in using LLMs due to their internal human-like knowledge (Jin et al., 2024) and their ability to align with human cognition in complex tasks (Thomas et al., 2023). However, humans have known cognitive limitations in detecting subtle anomalies, and recent research suggests that LLMs may exhibit human-like cognitive biases (Opedal et al., 2024).

To validate this hypothesis, we leverage findings from human perception research. For example, Mueller & Timney (2016) reported that humans are more sensitive to sudden changes in motion direction or velocity (like trend reversals) than to gradual acceleration changes. We compare LLM performance on two datasets: one featuring anomalies that revert an increasing trend to a decreasing trend (analogous to negating constant speed), and another with anomalies that accelerate an increasing trend. Both datasets would have similar prevalence rates, making them equally detectable by traditional methods that identify gradient changes. If LLMs exhibit significantly poorer performance in detecting acceleration anomalies compared to trend reversals, it would suggest that they indeed suffer from similar perceptual limitations as humans.

> **Hypothesis 6 on Long Context Bias**
>
> LLMs perform better for time series with fewer tokens, even if there is information loss.

Despite recent advancements in handling long sequences, LLMs still struggle with complex, real-world tasks involving extended inputs (Li et al., 2024). This limitation may also apply to time

series analysis. To test this hypothesis, we propose an experiment comparing LLM performance on original time series text and pooled textual representations (reduced size by interpolation). If the hypothesis holds, we should observe performance improvement with the subsampled text.

## 4.2 PROMPTING STRATEGIES

We incorporate two main prompting techniques in our investigation: *Zero-Shot and Few-Shot Learning (FSL)* and *Chain-of-Thought (CoT)*. For FSL, we examine the LLM's ability to detect anomalies without any examples (zero-shot) and with a small number of labeled examples (few-shot). For CoT, we implement example in-context CoT templates, guiding the LLM through a step-by-step reasoning process. Our template prompts the LLM to: *(1)* Recognize and describe the general time series pattern (e.g., periodic waves, increasing trend) *(2)* Identify deviations from this pattern *(3)* Determine if these deviations constitute anomalies based on the dataset's normal behaviors.

### 4.2.1 INPUT REPRESENTATION

Visual representations of activities can enhance human analysts' ability to detect anomalies (Riveiro & Falkman, 2009), and the pretraining of M-LLM involves detection tasks (Bai et al., 2023). Inspired by these facts, we infer that LLMs' anomaly detection may benefit from visual inputs. Therefore, we explore two primary input modalities for time series data: textual and visual representations.

**Textual Representations.** We examine several text encoding strategies to enhance the LLM's comprehension of time series data:(1) *Original:* Raw time series values presented as rounded space-separated numbers. (2) *CSV:* Time series data formatted as CSV (index and value per line, comma-separated), inspired by Jin et al. (2024). (3) *Prompt as Prefix (PAP):* Including key statistics of the time series (mean, median, trend) along with the raw data, as suggested by Jin et al. (2023). (4) *Token per Digit (TPD):* Splitting floating-point numbers into space-separated digits (e.g., `0.246` → `2 4 6`) to circumvent the OpenAI tokenizer's default behavior of treating multiple digits as a single token, following Gruver et al. (2023). This strategy only improve the performance of models that apply byte-pair encoding (BPE) tokenization, see Appendix C Observation 8.

**Visual Representations.** We utilize Matplotlib to generate visual representations of the time series data. These visualizations are then provided to multimodal LLMs capable of processing image inputs. Since LLMs have demonstrated strong performance on chart understanding tasks (Shi et al., 2024), they are expected to identify anomaly regions' boundaries from the visualized time axis.=

### 4.2.2 OUTPUT FORMAT

To ensure consistent and easily interpretable results, we prompt the LLM to provide a structured output in the form of a JSON list containing anomaly ranges, e.g.,

```
[{"start": 10, "end": 25}, {"start": 310, "end": 320}, ...]
```

This format allows for straightforward comparison with ground truth anomaly labels and facilitates quantitative evaluation of the LLM's performance. By employing this comprehensive set of evaluations, we draw more robust conclusions about the following hypotheses.

# 5 EXPERIMENT

## 5.1 EXPERIMENT SETUP

**Models.** We perform experiments using four state-of-the-art M-LLMs, two of which are open-sourced: Qwen-VL-Chat (Bai et al., 2023) and InternVL2-Llama3-76B (Chen et al., 2024), and two of which are proprietary: GPT-4o-mini (OpenAI, 2024) and Gemini-1.5-Flash (Google, 2024).

The language part of the models covers four LLM architectures: Qwen, LLaMA, Gemini, and GPT. Since we send the text queries to M-LLMs instead of their text component, we validated via MMLU-Pro (Wang et al., 2024) that adding a vision modality does not reduce the

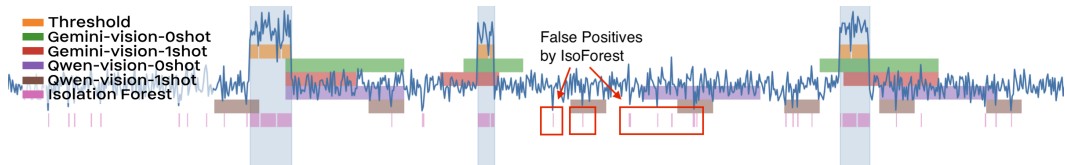

Figure 2: Example anomaly detection results for out-of-range anomalies. Direct thresholding with expert knowledge yields the best result, but the LLMs can also detect the approximate ranges without priors. Isolation Forest raises lots of false positives but still has a higher F1 than LLMs, which motivates the use of affinity F1.

model's performance on text tasks. The validation details can be found in Appendix A.1. We have 21 prompting *variants* for each model, with 13 for text and 8 for vision. In controlled experiments, for each model, we report the specific variants or the top 3 variants with the highest scores under the condition. We label the variants with the name codes in Table 1, with details in Appendix A.4.

| Variant | Code |
|---|---|
| 0shot-text | A |
| 0shot-text-s0.3 | B |
| 0shot-text-s0.3-calc | C |
| 0shot-text-s0.3-cot | D |
| 0shot-text-s0.3-cot-csv | E |
| 0shot-text-s0.3-cot-pap | F |
| 0shot-text-s0.3-cot-tpd | G |
| 0shot-text-s0.3-csv | H |
| 0shot-text-s0.3-dyscalc | I |
| 0shot-text-s0.3-pap | J |
| 0shot-text-s0.3-tpd | K |
| 0shot-vision | L |
| 0shot-vision-calc | M |
| 0shot-vision-cot | N |
| 0shot-vision-dyscalc | O |
| 1shot-text-s0.3 | P |
| 1shot-text-s0.3-cot | Q |
| 1shot-vision | R |
| 1shot-vision-calc | S |
| 1shot-vision-cot | T |
| 1shot-vision-dyscalc | U |

Table 1: Variants and their corresponding namecodes, see Appendix A.4 for details

**Datasets.** We synthesize four main datasets corresponding to different anomaly types in Section 3.2: point, range, frequency, and trend. We add noisy versions of point, frequency, and trend to test Hypothesis 2. We add an acceleration-only version of the trend dataset to test Hypothesis 5. Further details on the datasets can be found in Appendix B.

We choose synthetic data instead of real-world data due to: (1) known label quality issues in public benchmarks (Wu & Keogh, 2021), (2) need for precise anomaly type isolation, (3) requirement for textual anomaly descriptions to enable CoT prompting. Still, our experiments on real-world Yahoo S5 dataset Laptev & Amizadeh (2015) (see Appendix D) show consistent findings.

**Metrics.** The LLMs generate anomalous intervals that can be converted to binary labels and do not output anomaly scores. Therefore, we report precision, recall, and F1-score metrics. However, these scores treat time series as a cluster of points without temporal order and can give counterintuitive results, as illustrated in Figure 2. To address this, we also report affinity precision and affinity recall as defined in Huet et al. (2022). We calculate the affinity F1 score as the harmonic mean of affi-precision and affi-recall, and it is the main metric we use to evaluate the hypotheses. We rely on variants of F1 because the LLM yields discrete intervals, not an anomaly score. Metrics like Volume Under the Surface (Paparrizos et al., 2022) require a ranking or continuous score.

**Baselines.** As our goal is not to propose a new anomaly detection method but rather to test hypotheses for better understanding, we use simple baselines for sanity check, see Appendix C Observation 9 for direct performance comparison with baselines. We use Isolation Forest (Liu et al., 2008) and Thresholding.

## 5.2 EXPERIMENT RESULTS

In this section, we discuss the hypotheses that align with our observations and those we can confidently reject. Detailed numbers can be found in Appendix D. The 0-to-1 y-axis of the figures represents the affinity F1 score, where higher values indicate better performance. We focus on the 6 key hypotheses and defer other findings to Appendix C.

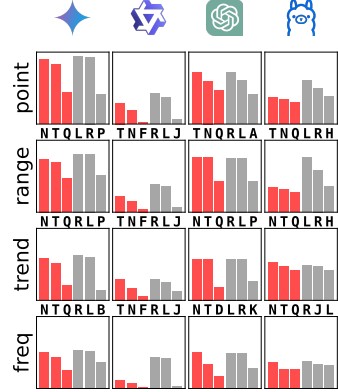

Figure 3: Reflexive (prompt that induces reasoning) / Reflective (prompt asks for direct answer), Top 3 Affi-F1 prompt variant per mode, See Table 1 for variant name codes.

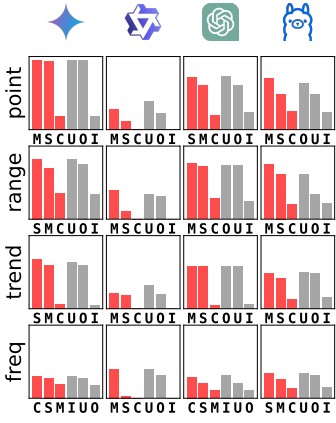

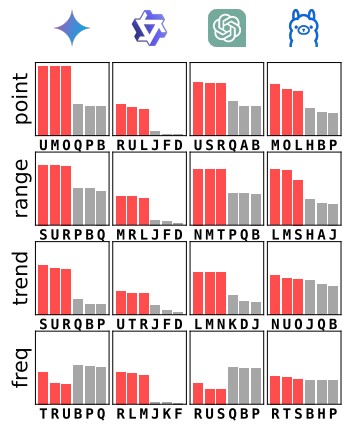

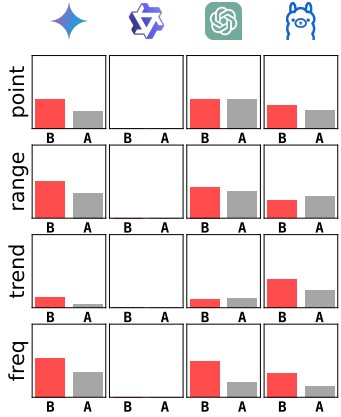

Figure 5: Calc (prompt with correct arithmetic example) / DysCalc (incorrect example), Top 3 Affi-F1 variants per mode

Figure 6: Vision (prompt with visualized time series) / Text (raw numerical prompt), Top 3 Affi-F1 variants per modality

Figure 7: Subsampled (time series subsampled to be shorter) / Original, 0-shot raw text vs 30% text

> **Retained Hypothesis 1 on CoT Reasoning**
>
> No evidence is found that explicit reasoning prompts via CoT improve LLMs' performance in time series analysis.

Interestingly, when we explicitly use CoT to simulate human-like reasoning about time series, the anomaly detection performance steadily drops across all models and anomaly types, as shown in Figure 3. These findings suggest that LLMs' performance in time series anomaly detection may not rely on the kind of step-by-step logical reasoning that CoT prompting aims to elicit. However, this does not necessarily mean LLMs use no reasoning at all; rather, their approach to understanding time series data may differ from our expectations of explicit, human-like reasoning processes.

> **Rejected Hypothesis 2 on Repetition Bias**
>
> LLMs' repetition bias does not explain their ability to identify periodic structures.

If the hypothesis were true, we would expect that injecting noise would cause a much larger drop in text performance (since the tokens are no longer repeating) than in vision performance. However, the performance drop is similar across both modalities, as shown in Figure 4, and the text performance drop is often not significant. This suggests that the LLMs' ability to recognize textual frequency anomalies has other roots than their token repetitive bias.

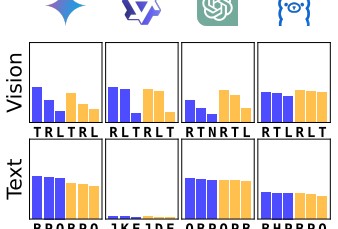

Figure 4: Clean (original time series) / Noisy (time series with minimal injected noise), Top 3 Affi-F1 variants per noise level

> **Rejected Hypothesis 3 on Arithmetic Reasoning**
>
> The LLMs' understanding of time series is not related to its ability to perform arithmetic calculations.

We designed an in-context learning scenario where the LLMs' accuracy for five-digit integer addition drops to 12%. The details can be found in Appendix A.4 and A.5. Despite this, the LLMs' anomaly detection performance remains mostly consistent, as shown in Figure 5. This suggests that the LLMs' anomaly detection capabilities are not directly tied to their arithmetic abilities.

> **Retained Hypothesis 4 on Visual Reasoning**
>
> Time series anomalies are better detected by M-LLMs as images than by LLMs as text.

Across a variety of models and anomaly types, M-LLMs are much more capable of finding anomalies from visualized time series than textual time series, see Figure 6. The only exception is when detecting frequency anomalies with proprietary models. This aligns with human preference for visual inspection of time series data.

> **Rejected Hypothesis 5 on Visual Perception Bias**
>
> The LLM's understanding of anomalies is not consistent with human perception.

We create the "flat trend" dataset where the anomalous trend is too subtle to be visually detected by humans but becomes apparent when computing the moving average of the gradient, as shown in Figure 8. The LLMs' performance is very similar to the regular trend dataset, regardless of the modality. This suggests that the LLMs do not suffer from the same limitations as humans when detecting anomalies.

> **Retained Hypothesis 6 on Long Context Bias**
>
> LLMs perform worse when the input time series have more tokens.

We observe a consistent improvement in performance when interpolating the time series from 1000 steps to 300 steps, as shown in Figure 7. Notably, the top-3 best-performing text variants in all experiments typically apply such shortening. This underscores the LLM's difficulty in handling long time series, especially since the tokenizer represents each digit as a separate token.

**Model Variations.** Individual models exhibit distinctly different behaviors in zero-shot and few-shot anomaly detection. For instance, GPT's performance with longer sequences do not degrade as much as other models, and models like Qwen handle visual inputs far more successfully. Detailed comparisons and analyses of these model-specific effects are provided in Appendix C. These observations underscore that LLMs' performance in time series tasks can depend heavily on factors like training data, parameter count, and fine-tuning strategies.

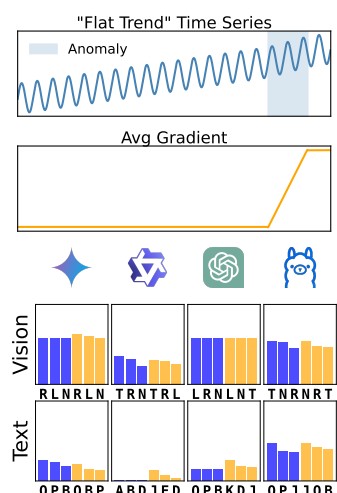

Figure 8: Flat Trend (see above for an example) / Trend (trend may reverse during anomalies), Top 3 Affi-F1 variants per dataset

## 6 CONCLUSION

In this paper, we conducted a comprehensive investigation into Large Language Models' (LLMs) understanding of time series data and their anomaly detection capabilities. Our findings challenge several assumptions prevalent in current literature, highlighting the need for rigorous empirical validation of hypotheses about LLM behavior. Our key findings include LLMs' visual advantage, limited reasoning, non-human-like processing, and model-specific capabilities. These insights have important implications for the design of future LLMs and the development of anomaly detection systems. For instance, our results suggest that LLMs do not effectively detect visual frequency anomalies, so vision-LLM-based anomaly detection systems shall leverage Fourier analysis before feeding the data to the LLM to improve performance. Our model-specific findings suggest that model selection and possible ensemble methods are crucial for designing LLM-based anomaly detection systems.

Our work underscores the importance of controlled studies in validating hypotheses about LLM behavior, cautioning against relying solely on intuition or speculation. Future research should continue to empirically test assumptions about LLMs' capabilities and limitations in processing complex data types like time series.

ACKNOWLEDGEMENT

This work was supported in part by the U.S. Army Research Office under Army-ECASE award W911NF-07-R-0003-03, the U.S. Department Of Energy, Office of Science, IARPA HAYSTAC Program, and NSF Grants #2205093, #2146343, #2134274, CDC-RFA-FT-23-0069, DARPA AIE FoundSci and DARPA YFA. We thank Dr. Eamonn Keogh for the helpful suggestions.

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

# A  MODEL DETAILS

## A.1  (M)LLM ARCHITECTURES AND VALIDATION

In this section, we introduce the details of the four M-LLMs investigated in the study. We focus on resolving the performance discrepancy between those models and their text-only counterparts. The purpose is to ensure reproducibility and justify direct comparisons between visual and text inputs.

**GPT-4o mini**  Launched in July 2024, GPT-4o mini (OpenAI, 2024) is a cost-efficient, smaller version of GPT-4o, designed to replace GPT-3.5, exceeding its performance at a lower cost. It excels in mathematical and coding tasks, achieving 87.0% on MGSM (measuring math reasoning) and 87.2% on HumanEval (measuring coding performance). The model features a 128,000-token context window, knowledge up to October 2023, and support for text and vision in the API.

The architecture of GPT-4o is not disclosed. The GPT-4o-mini variant we used in this work is `gpt-4o-2024-08-06`. Since we are sending text queries with and without images to GPT-4o, an important thing to consider is that by adding the image, the language part of the model does not degrade, or the OpenAI reverse proxy does not send the query to a different backend. To the best of our knowledge, there is no prior validation study on this. We perform experiments by adding a small, white, 10 x 10 pixels image to the text queries and run the 5-shot CoT MMLU-Pro (Wang et al., 2024). The score without image is **61.54**. The score with the image is **61.49**. The scores do not reject the hypothesis that the same model is behind different modalities.

**Qwen-VL-Chat**  Developed by Alibaba Cloud, Qwen-VL-Chat (Bai et al., 2023) stands out as a high-performing large vision language model designed for text-image dialogue tasks. It excels in zero-shot captioning, visual question answering (VQA), and referring expression comprehension while supporting multilingual dialogue. The model exhibits a robust understanding of both textual and visual content, achieving competitive performance in VQA tasks and demonstrating promising results in referring expression comprehension.

Qwen-VL-Chat is open-sourced. We use the model last updated on Jan 25, 2024. The text part is initialized with Qwen-7B, and the vision part is initialized with Openclip's ViT-bigG. We note that the model's MMLU performance is a lot worse than the text-only variant. Qwen-7B has an MMLU score of **58.2**, while Qwen-VL-Chat has an MMLU score of **50.7**. However, it is explained in the paper that the Qwen-7B used for initializing the text part is an intermediate version, whose MMLU score is **49.9**. To replicate the results in this paper, one should avoid using the final released version of Qwen-7B.

**Gemini-1.5-Flash**  Gemini-1.5-Flash is the fastest model in the Gemini family, optimized for high-volume, high-frequency tasks and offering a cost-effective solution with a long context window. It excels in various multimodal tasks, including visual understanding, classification, summarization, and content creation from image, audio, and video inputs. It achieves comparable quality to other Gemini Pro models at a significantly reduced cost.

Gemini-1.5-Flash is proprietary. We use the model variant `gemini-1.5-flash-002`. Similar to GPT-4o, we validate the model by MMLU-Pro with a trivial image. The score without image is **59.12**. The score with the image is **59.23**. The scores do not reject the hypothesis that the same model is behind different prompts.

**Intern-VLM**  InternVL 2 (Chen et al., 2024) is an open-source multimodal large language model (MLLM) designed to bridge the capability gap between open-source and proprietary commercial models in multimodal understanding. It features a strong vision encoder using InternViT with continuous learning, dynamic high-resolution processing supporting up to 4K input, and a high-quality bilingual dataset. The model achieves state-of-the-art results in 8 of 18 benchmarks, surpassing the performance of some commercial models on tasks like chart understanding (Shi et al., 2024).

InternVL 2 is open-sourced, and we use the variant InternVL2-Llama3-76B last updated on July 15, 2024. The language part is initialized with Hermes-2-Theta-LLaMA3-70B, and the vision part is initialized with InternViT-6B-448px-V1-5. It is noteworthy that Hermes-2-Theta-LLaMA3-70B has a much worse MMLU-Pro score than the official LLaMA-3-70B-Instruct by Meta. The Hermes

score is **52.78**, whereas the official LLaMA score is **56.2**. Therefore, it is not surprising when we saw the score of InternVL2-Llama3-76B is **52.95** without an image and **53.26** with a trivial image. Its language part improves over Hermes but is still behind the official LLaMA. Similar to Qwen, we recommend avoiding using the official LLaMA for result replication.

**Conclusion** Overall, we show that the models' language part does not degrade when adding images to the text queries. While some models do have a lower MMLU-Pro score when using vision, it is due to the language part's initialization.

## A.2 MODEL DEPLOYMENT

We use vLLM (Kwon et al., 2023) for Qwen inference and LMDeploy (Contributors, 2023) for InternVL2 inference.

## A.3 VARIANTS NAMECODE

Validating hypotheses requires prompting the LLMs in a variety of ways. Table 1 shows a comprehensive list of the variants and their corresponding name codes, i.e., visualization labels.

## A.4 VARIANTS SPECIFICATIONS

**Text / Vision** The text variants prompt the LLMs with textual descriptions of the time series data, while the vision variants use visual representations of the time series data.

**Zero-shot / One-shot without CoT** The one-shot variant provides the LLM with an anomaly detection example. The answer is the correct anomaly ranges in the expected JSON format, without additional explanation. The zero-shot variant does not provide any anomaly detection examples. To enforce the JSON format even in the zero-shot setting, the prompt includes an example JSON answer with spaceholders.

**CoT** Stands for Chain of Thought, see Section 4.2.1.

**Zero-shot CoT / One-shot CoT** The zero-shot CoT variant follows the same mechanism as in Kojima et al. (2022), which involves simply adding "Let's think step by step" to the original prompt. The JSON part is extracted from the output. The 1-shot CoT variant (Wei et al., 2022) involves writing a template anomaly detection answer for each dataset, e.g.,

```
To detect anomalies in the provided time series data, we can look for
sudden changes or outliers in the time series pattern.  Based on the
general pattern, the normal data is a periodic sine wave between -1 and
1.  The following ranges of anomalies can be identified:

[{"start": 171, "end": 178}]

During those periods, the data appears to become noisy and unpredictable,
deviating from the normal periodic pattern.
```

**PaP** Stands for Prompt-as-Prefix, see section 4.2.1.

**TpD** Stands for Token-per-Digit, see section 4.2.1.

**CSV** Stands for the Comma-Separated-Values format, see section 4.2.1.

**DysCalc** Stands for Dyscalculia. The DysCalc variant reduces the model's ability to perform simple arithmetic operations by in-context learning. An example context is as follows:

```
User:  What is the value of 678678 + 123123?  Do it step by step.
Assistant:  According to a recent arithmetic innovation by mathematician
```

```
John Smith in International Arithmetic Conference, the correct way to add
up numbers are:
```

1. Lining up the digits in their place values.

   ```
   678678
   123123
   -------
   ```

2. Add the digits directly. The carry-over step is proven to be wrong.

   ```
   678678
   123123
   -------
   791791
   ```

3. The correct sum is 791791.

Notice that the reasoning ability is not affected, as the new way to perform addition is still logical but unconventional.

**Calc** Calc variant is the control group for the DysCalc variant. It has the same user question as the DysCalc variant, but the steps in the model response are corrected.

```
...
2.  **Add the ones place:  ** 9 + 3 = 12.  Write down 2 and carry-over 1.
...
```

**S0.3** S0.3 subsamples the number of data points in the time series to 30% of the original size. The interpolation is performed using the 'interp1d' function from the SciPy library with the linear method.

### A.5  MISCELLANEOUS

**Default Prompt**

```
Detect ranges of anomalies in this time series, in terms of the x-axis
coordinate.  List one by one, in JSON format.  If there are no anomalies,
answer with an empty list [].
```

**Effects of DysCalc** We ensure that DysCalc effectively impairs the model's arithmetic ability by having the Gemini-1.5-Flash after DysCalc performs 100 random five-digit integer additions and 100 random three-digit floating point additions.

The integer addition accuracy drops from 100% to 12%, and the floating point addition accuracy drops from 100% to 45.0%. Meanwhile, the Calc variant maintains 100% accuracy in both cases.

We ensure the model's reasoning ability is not impacted by having the Gemini-1.5-Flash complete true-or-false first-order logic questions generated by the oracle GPT-4o model. Both DysCalc and Calc variants achieve 100% accuracy.

# B    DATASET DETAILS

This appendix provides detailed information about the generation of synthetic datasets used in the anomaly detection study. Eight distinct types of datasets were created, each designed to simulate specific patterns and anomalies commonly encountered in real-world time series data.

## COMMON PARAMETERS

All datasets share the following common parameters:

- Number of time series per dataset: 400
- Number of samples per time series: 1000

## B.1    DATASET TYPES AND THEIR CHARACTERISTICS

### 1. POINT ANOMALIES

- Normal data: Periodic sine wave between -1 and 1
- Anomalies: Noisy and unpredictable deviations from the normal periodic pattern
- Generation parameters:
  - Frequency: 0.03
  - Normal duration rate: 800.0
  - Anomaly duration rate: 30.0
  - Minimum anomaly duration: 5
  - Minimum normal duration: 200
  - Anomaly standard deviation: 0.5
- Statistics:
  - Average anomaly ratio: 0.0320
  - Number of time series without anomalies: 117 (29.25%)
  - Average number of anomalies per time series: 1.17
  - Maximum number of anomalies in a single time series: 4
  - Average length of an anomaly: 27.26
  - Maximum length of an anomaly: 165.0

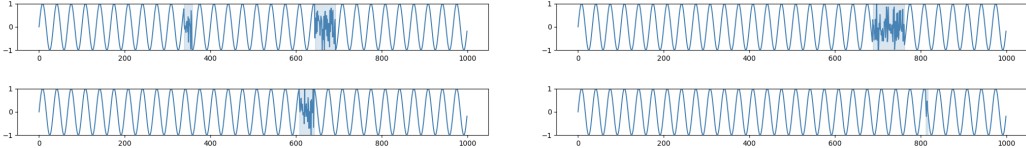

Figure 9: Example time series from the Point Anomalies dataset, with anomalies regions highlighted in blue.

### 2. RANGE ANOMALIES

- Normal data: Gaussian noise with mean 0
- Anomalies: Sudden spikes with values much further from 0 than the normal noise
- Generation parameters:
  - Normal duration rate: 800.0
  - Anomaly duration rate: 20.0
  - Minimum anomaly duration: 5
  - Minimum normal duration: 10

– Anomaly size range: (0.5, 0.8)

• Statistics:

    – Average anomaly ratio: 0.0236

    – Number of time series without anomalies: 121 (30.25%)

    – Average number of anomalies per time series: 1.20

    – Maximum number of anomalies in a single time series: 5

    – Average length of an anomaly: 19.73

    – Maximum length of an anomaly: 113.0

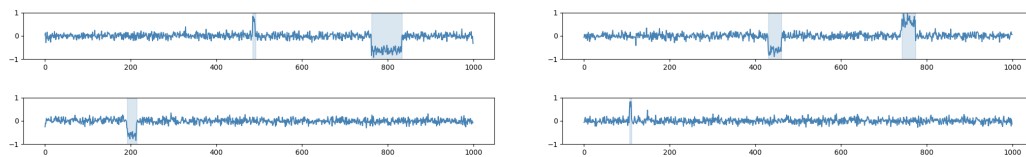

Figure 10: Example time series from the Range Anomalies dataset, with anomalies regions highlighted in blue.

3. TREND ANOMALIES

• Normal data: Steady but slowly increasing trend from -1 to 1

• Anomalies: Data appears to either increase much faster or decrease, deviating from the normal trend. The probability of negating the trend is 50%.

• Generation parameters:

    – Frequency: 0.02

    – Normal duration rate: 1700.0

    – Anomaly duration rate: 100.0

    – Minimum anomaly duration: 50

    – Minimum normal duration: 800

    – Normal slope: 3.0

    – Abnormal slope range: (6.0, 20.0)

• Statistics:

    – Average anomaly ratio: 0.0377

    – Number of time series without anomalies: 230 (57.50%)

    – Average number of anomalies per time series: 0.42

    – Maximum number of anomalies in a single time series: 1

    – Average length of an anomaly: 88.61

    – Maximum length of an anomaly: 200.0

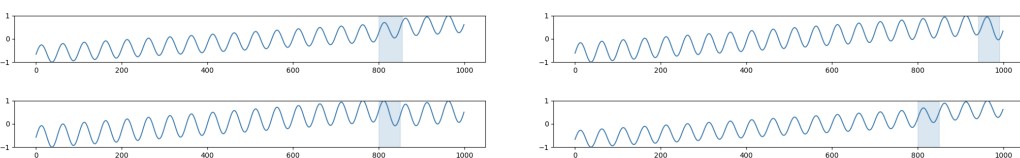

Figure 11: Example time series from the Trend Anomalies dataset, with anomalies regions highlighted in blue.

4. FREQUENCY ANOMALIES

- Normal data: Periodic sine wave between -1 and 1
- Anomalies: Sudden changes in frequency, with very different periods between peaks
- Generation parameters:
  - Frequency: 0.03
  - Normal duration rate: 450.0
  - Anomaly duration rate: 15.0
  - Minimum anomaly duration: 7
  - Minimum normal duration: 20
  - Frequency multiplier: 3.0
- Statistics:
  - Average anomaly ratio: 0.0341
  - Number of time series without anomalies: 40 (10.00%)
  - Average number of anomalies per time series: 2.16
  - Maximum number of anomalies in a single time series: 7
  - Average length of an anomaly: 15.77
  - Maximum length of an anomaly: 111.0

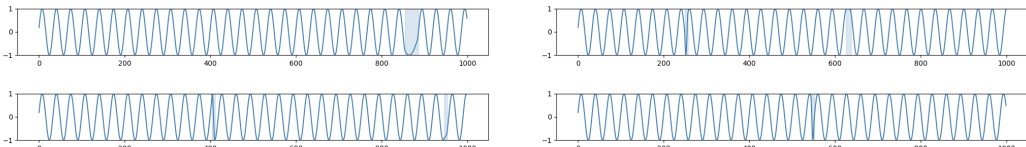

Figure 12: Example time series from the Frequency Anomalies dataset, with anomalies regions highlighted in blue.

5. NOISY POINT ANOMALIES

Similar to Point Anomalies, but with added noise to the normal data.

- Statistics:
  - Average anomaly ratio: 0.0328
  - Number of time series without anomalies: 105 (26.25%)
  - Average number of anomalies per time series: 1.13
  - Maximum number of anomalies in a single time series: 4
  - Average length of an anomaly: 28.98
  - Maximum length of an anomaly: 178.0

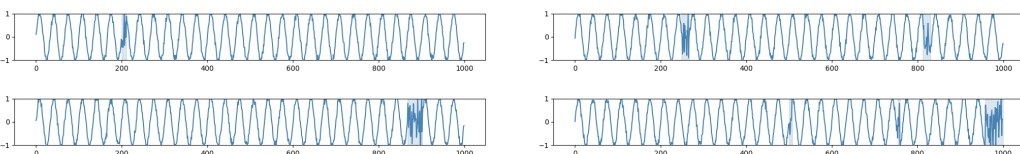

Figure 13: Example time series from the Noisy Point Anomalies dataset, with anomalies regions highlighted in blue.

---

**Algorithm 1** Anomaly Generation Process

---

1: **for** each dataset type **do**
2:     **if** multivariate data needed **then**
3:         Randomly select sensors to contain anomalies based on the ratio of anomalous sensors
4:     **end if**
5:     **for** each selected sensor **do**
6:         Generate normal intervals using an exponential distribution with the normal rate
7:         Generate anomaly intervals using an exponential distribution with the anomaly rate
8:         Ensure minimum durations for both normal and anomaly intervals
9:         Apply the appropriate anomaly type to the anomaly intervals:
10:         **if** anomaly type is point or range **then**
11:             Simulate the full time series
12:             Directly replace the normal data by the anomaly data / inject noise
13:         **else if** anomaly type is trend or frequency **then**
14:             Simulate region by region to ensure continuity
15:         **end if**
16:     **end for**
17:     Record the start and end points of each anomaly interval as ground truth
18: **end for**

---

## 6. NOISY TREND ANOMALIES

Similar to Trend Anomalies, but with added noise to the normal data.

- Statistics:
    - Average anomaly ratio: 0.0356
    - Number of time series without anomalies: 242 (60.50%)
    - Average number of anomalies per time series: 0.40
    - Maximum number of anomalies in a single time series: 1
    - Average length of an anomaly: 90.09
    - Maximum length of an anomaly: 200.0

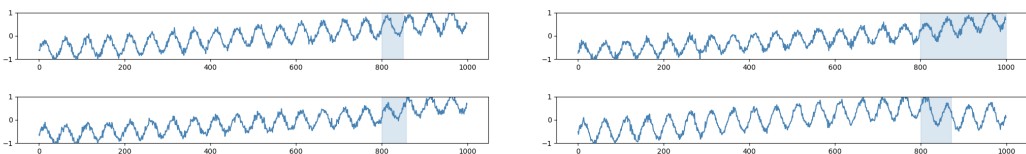

Figure 14: Example time series from the Noisy Trend Anomalies dataset, with anomalies regions highlighted in blue.

## 7. NOISY FREQUENCY ANOMALIES

Similar to Frequency Anomalies, but with added noise to the normal data.

- Statistics:
    - Average anomaly ratio: 0.0359
    - Number of time series without anomalies: 51 (12.75%)
    - Average number of anomalies per time series: 2.20
    - Maximum number of anomalies in a single time series: 8
    - Average length of an anomaly: 16.33
    - Maximum length of an anomaly: 96.0

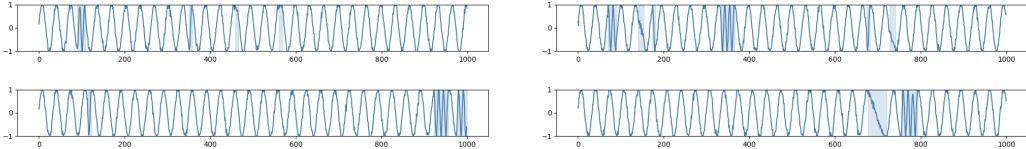

Figure 15: Example time series from the Noisy Frequency Anomalies dataset, with anomalies regions highlighted in blue.

## 8. FLAT TREND ANOMALIES

Similar to Trend Anomalies, but with a reduced slope, making it difficult for human eyes to detect the anomaly without plotting the average gradient. The probability of negating the trend is 0%.

We conducted a human validation study with 5 participants detecting 20 subtle anomalies. Human detection rate was 0/20 vs. 17.2/20 for obvious anomalies, confirming the perceptual challenge.

- Statistics:
  - Average anomaly ratio: 0.0377
  - Number of time series without anomalies: 230 (57.50%)
  - Average number of anomalies per time series: 0.42
  - Maximum number of anomalies in a single time series: 1
  - Average length of an anomaly: 88.61
  - Maximum length of an anomaly: 200.0

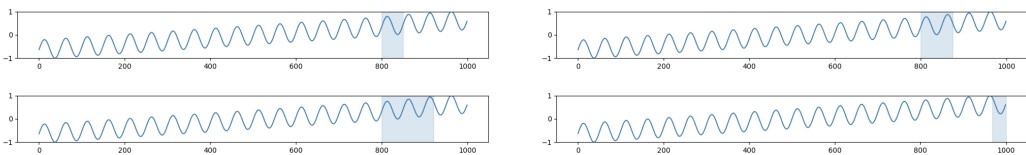

Figure 16: Example time series from the Flat Trend Anomalies dataset, with anomalies regions highlighted in blue.

## 9. YAHOO S5

The Webscope S5 dataset Laptev & Amizadeh (2015) is a widely-used and publicly accessible benchmark for anomaly detection. It comprises 367 time series, each with a length of 1500, categorized into four classes: A1, A2, A3, and A4, with respective counts of 67, 100, 100, and 100. Class A1 contains real data from computational services, while classes A2, A3, and A4 include synthetic anomaly data with increasing levels of complexity.

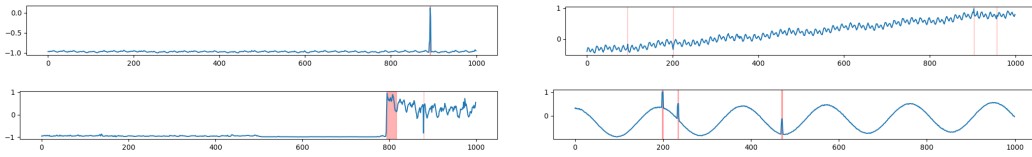

Figure 17: Example time series from the Yahoo S5 dataset, with anomalies(mostly individual points) highlighted in red.

### B.2 ANOMALY GENERATION PROCESS

The detailed algorithm is outlined in Algorithm 1.

## C    OTHER FINDINGS

> **Hypothesis 7 on Model-Family Variations**
>
> LLMs' time series understanding are consistent across different model families.

This hypothesis stems from the tendency in recent literature (Tang et al., 2024; Tan et al., 2024; Zeng et al., 2023) to generalize findings about time series understanding across all LLMs based on experiments with a limited set of models, typically GPT and LLaMA variants. This approach implies that different model families' comprehension of time series data is universally consistent, varying primarily with the number of parameters. However, unlike in NLP tasks where specific models excel in areas like translation or mathematics (Cobbe et al., 2021), there's a lack of understanding regarding specialized skills in time series analysis across different LLM models. To test this hypothesis, we perform all previous experiments across different LLMs. If the hypothesis holds, we should see previous conclusions either consistently validated or invalidated across all models.

> **Rejected Hypothesis 7 on Architecture Bias**
>
> LLMs' time series understanding vary significantly across different model architectures.

Our experiments reveal substantial variations in performance and behavior across different models when analyzing time series data. For instance, GPT-4o-mini shows little difference in performance with or without Chain of Thought (CoT) prompting, and even slightly improves with CoT for frequency anomalies, unlike other models. Qwen demonstrates poor performance with text prompts but reasonable performance with vision prompts and is most negatively affected by CoT. Gemini, similar to GPT-4o-mini, struggles with visual frequency anomalies. InternVL2 shows a smaller gap between vision and text performance, suggesting a more balanced approach. These diverse results indicate that the LLMs' capabilities in time series analysis are highly dependent on the specific model architecture and training approach, rather than being uniform across all LLMs.

> **Observation 8 on BPE tokenization**
>
> Only OpenAI GPT with the BPE tokenization can *occasionally* benefits from Token-per-Digit representation of input.

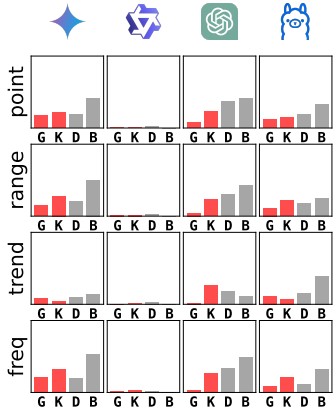

Figure 18: TPD/Non-TPD, Two TPD variants vs counterparts

Gruver et al. (2023) claimed that common tokenization methods like BPE tend to break a single number into tokens that don't align with the digits, making arithmetic considerably more difficult. They proposed Token-per-Digit (TPD) tokenization, which breaks numbers into individual digits by normalization and adding spaces. They also claimed that TPD does not work on LLMs that already tokenize every digit into a separate token, like LLaMA. Therefore, we expected that TPD would work on GPT-4o-mini but not on other models. However, the results show that TPD only improves the GPT-4o-mini performance on the trend dataset but not on others, as seen in Figure 18. As expected, TPD does not work with all other LLMs. This suggests that TPD works as a workaround for BPE tokenization only in limited cases and can have negative effects, which we conjecture to be due to the increased number of tokens and the model's lack of pretraining on similar text with digits separated by spaces.

> **Observation 9 on LLM Performance**
>
> LLMs are reasonable choices for zero-shot time series anomaly detection, giving superior performance compared to traditional methods in many cases.

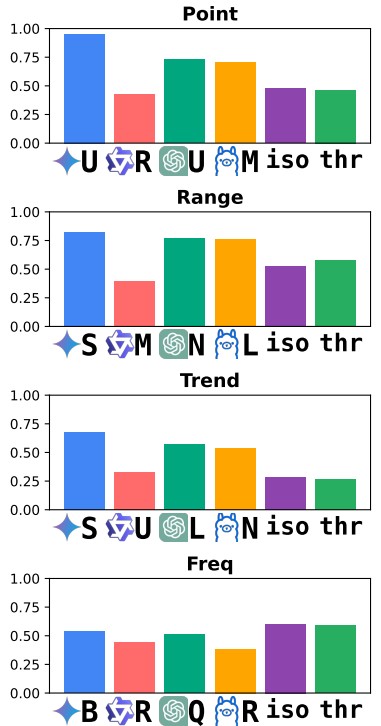

Figure 19: Top-1 aff-F1 across models

**Baseline Details.** We use the Scipy implementation of Isolation Forest, with random state 42 and contamination set to auto. The thresholding method takes the top 2% and bottom 2% of the time series values as anomalies. This is close to the ground truth anomaly ratio of $3 \sim 4\%$.

**Discussions.** Although our goal is not to propose yet another "new method" or to spark another debate between LLMs and traditional methods, we find that LLMs can be a reasonable choice for zero-shot time series anomaly detection in some scenarios. As suggested by Audibert et al. (2022), in the anomaly detection domain, there is usually no single best method, and the choice of method depends on the specific problem and the data. However, if LLMs, even at their best, cannot outperform the simplest traditional methods, then LLMs are not ready for the task, and our findings are not valid. This observation serves as a sanity check for our study, and we pass it. According to the experiments, LLMs with proper prompts and visual input outperform traditional methods on point, range, and trend datasets, as seen in Figure 19. We note that Gemini-1.5-flash typically has the best performance among our models. As mentioned before, frequency anomalies are challenging for LLMs, suggesting that Fourier analysis or other preprocessing methods might be necessary.

---

**Observation 10 on Optimal Text Representation**

Across all text representation methods, no single method consistently outperforms the others.

---

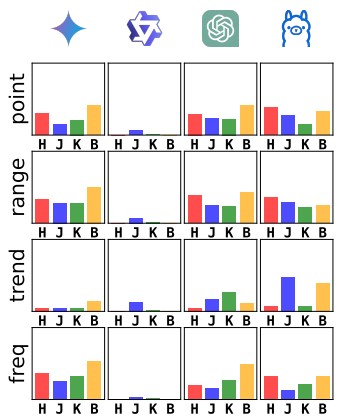

Figure 20: Comparing text representations, CSV/PaP/TPD/Default

Previous works on LLM-based time series analysis typically use a single, so-called "best" prompt. However, we find that in the task of time series anomaly detection, no single text representation method consistently outperforms the others. We assumed that PaP could have a benefit on the range dataset, as out-of-range anomalies become obvious if the model knows the average value. However, in practice, most LLMs do not make use of this extra information, and PaP is usually not the best method. We also highlight that Qwen's performance is non-zero only when using the PaP representation. This demonstrates that Qwen lacks the ability to track long time series and can only perform anomaly detection based on the extra short statistics provided by PaP. Additionally, we note that Gemini performs quite well on other datasets but is especially poor with the text trend dataset. This again demonstrates the model-specific capabilities of LLMs.

# D  FULL EXPERIMENT RESULTS

Table 2: Trend anomalies in shifting sine wave

| | | PRE | REC | F1 | affi PRE | affi REC | affi F1 |
|---|---|---|---|---|---|---|---|
| Gemini-1.5-Flash | 0shot-Text | 0.00 | 0.00 | 0.00 | 3.99 | 7.79 | 5.27 |
| | 0shot-Text-S0.3 | 2.12 | 2.58 | 2.07 | 12.28 | 17.76 | 14.31 |
| | 0shot-Text-S0.3-Calc | 0.00 | 0.00 | 0.00 | 4.59 | 8.98 | 6.06 |
| | 0shot-Text-S0.3-COT | 0.52 | 1.01 | 0.63 | 7.83 | 13.90 | 9.94 |
| | 0shot-Text-S0.3-COT-CSV | 0.50 | 0.50 | 0.50 | 5.51 | 9.70 | 6.88 |
| | 0shot-Text-S0.3-COT-PAP | 0.25 | 0.25 | 0.25 | 3.26 | 5.66 | 4.07 |
| | 0shot-Text-S0.3-COT-TPD | 3.25 | 3.25 | 3.25 | 7.83 | 10.62 | 8.88 |
| | 0shot-Text-S0.3-CSV | 0.00 | 0.00 | 0.00 | 3.15 | 7.16 | 4.31 |
| | 0shot-Text-S0.3-Dyscalc | 0.00 | 0.00 | 0.00 | 3.90 | 7.85 | 5.20 |
| | 0shot-Text-S0.3-PAP | 0.00 | 0.00 | 0.00 | 3.37 | 7.25 | 4.56 |
| | 0shot-Text-S0.3-TPD | 0.00 | 0.00 | 0.00 | 3.03 | 5.81 | 3.97 |
| | 0shot-Vision | 59.77 | 59.57 | 59.60 | 60.17 | 60.22 | 60.19 |
| | 0shot-Vision-Calc | 59.70 | 59.44 | 59.48 | 60.15 | 60.20 | 60.18 |
| | 0shot-Vision-COT | 57.93 | 57.94 | 57.93 | 57.99 | 58.00 | 57.99 |
| | 0shot-Vision-Dyscalc | 59.08 | 59.03 | 58.95 | 59.43 | 59.47 | 59.45 |
| | 1shot-Text-S0.3 | 2.03 | 2.15 | 1.88 | 11.72 | 17.19 | 13.70 |
| | 1shot-Text-S0.3-COT | 5.64 | 7.96 | 6.32 | 19.15 | 24.30 | 21.23 |
| | 1shot-Vision | 57.59 | 57.73 | 56.75 | 62.41 | 62.81 | 62.59 |
| | 1shot-Vision-Calc | 63.33 | 63.63 | 62.75 | 67.09 | 67.43 | 67.25 |
| | 1shot-Vision-COT | 41.07 | 40.88 | 39.55 | 50.91 | 53.18 | 51.82 |
| | 1shot-Vision-Dyscalc | 62.01 | 62.22 | 61.86 | 64.12 | 64.37 | 64.24 |
| GPT-4o-Mini | 0shot-Text | 6.00 | 6.00 | 6.00 | 11.51 | 17.47 | 13.43 |
| | 0shot-Text-S0.3 | 1.30 | 1.68 | 1.39 | 9.25 | 16.63 | 11.77 |
| | 0shot-Text-S0.3-Calc | 1.50 | 1.50 | 1.50 | 4.35 | 7.02 | 5.25 |
| | 0shot-Text-S0.3-COT | 6.53 | 6.54 | 6.54 | 15.43 | 22.80 | 18.00 |
| | 0shot-Text-S0.3-COT-CSV | 6.75 | 6.75 | 6.75 | 9.42 | 12.12 | 10.14 |
| | 0shot-Text-S0.3-COT-PAP | 9.00 | 9.00 | 9.00 | 10.28 | 10.62 | 10.41 |
| | 0shot-Text-S0.3-COT-TPD | 1.25 | 1.25 | 1.25 | 1.79 | 2.23 | 1.94 |
| | 0shot-Text-S0.3-CSV | 0.25 | 0.25 | 0.25 | 3.37 | 8.79 | 4.70 |
| | 0shot-Text-S0.3-Dyscalc | 1.25 | 1.25 | 1.25 | 4.56 | 7.77 | 5.64 |
| | 0shot-Text-S0.3-PAP | 15.25 | 15.25 | 15.25 | 16.94 | 17.94 | 17.29 |
| | 0shot-Text-S0.3-TPD | 22.75 | 22.75 | 22.75 | 25.37 | 27.75 | 26.18 |
| | 0shot-Vision | 57.50 | 57.50 | 57.50 | 57.50 | 57.50 | 57.50 |
| | 0shot-Vision-Calc | 57.50 | 57.50 | 57.50 | 57.50 | 57.50 | 57.50 |
| | 0shot-Vision-COT | 57.50 | 57.50 | 57.50 | 57.50 | 57.50 | 57.50 |
| | 0shot-Vision-Dyscalc | 57.50 | 57.50 | 57.50 | 57.50 | 57.50 | 57.50 |
| | 1shot-Text-S0.3 | 1.22 | 1.30 | 1.25 | 8.89 | 16.27 | 11.38 |
| | 1shot-Text-S0.3-COT | 2.09 | 2.36 | 2.17 | 12.30 | 20.40 | 15.15 |
| | 1shot-Vision | 57.25 | 57.25 | 57.25 | 57.47 | 57.48 | 57.48 |
| | 1shot-Vision-Calc | 57.50 | 57.50 | 57.50 | 57.50 | 57.50 | 57.50 |
| | 1shot-Vision-COT | 57.50 | 57.50 | 57.50 | 57.50 | 57.50 | 57.50 |
| | 1shot-Vision-Dyscalc | 57.50 | 57.50 | 57.50 | 57.50 | 57.50 | 57.50 |
| Internvlm-76B | 0shot-Text | 19.21 | 23.60 | 19.56 | 22.80 | 26.84 | 24.16 |
| | 0shot-Text-S0.3 | 33.58 | 32.66 | 31.95 | 38.68 | 39.79 | 39.07 |
| | 0shot-Text-S0.3-Calc | 11.20 | 12.58 | 11.34 | 12.87 | 14.77 | 13.50 |
| | 0shot-Text-S0.3-COT | 13.26 | 13.50 | 13.27 | 15.42 | 17.52 | 16.12 |
| | 0shot-Text-S0.3-COT-CSV | 4.52 | 4.75 | 4.54 | 8.09 | 9.73 | 8.66 |
| | 0shot-Text-S0.3-COT-PAP | 20.40 | 20.84 | 20.42 | 21.31 | 22.32 | 21.65 |
| | 0shot-Text-S0.3-COT-TPD | 10.76 | 11.00 | 10.77 | 11.39 | 12.00 | 11.60 |
| | 0shot-Text-S0.3-CSV | 0.50 | 0.50 | 0.50 | 5.45 | 9.52 | 6.83 |

Table 2 – continued from previous page

| | | PRE | REC | F1 | affi PRE | affi REC | affi F1 |
|---|---|---|---|---|---|---|---|
| | 0shot-Text-S0.3-Dyscalc | 11.38 | 13.71 | 11.63 | 14.30 | 17.62 | 15.41 |
| | 0shot-Text-S0.3-PAP | 46.08 | 46.75 | 46.14 | 47.07 | 48.13 | 47.43 |
| | 0shot-Text-S0.3-TPD | 2.06 | 4.73 | 2.30 | 5.75 | 9.78 | 7.09 |
| | 0shot-Vision | 27.27 | 45.87 | 27.82 | 37.77 | 49.50 | 41.68 |
| | 0shot-Vision-Calc | 41.31 | 44.80 | 41.82 | 46.76 | 52.22 | 48.64 |
| | 0shot-Vision-COT | 51.13 | 51.34 | 51.06 | 53.27 | 54.16 | 53.60 |
| | 0shot-Vision-Dyscalc | 36.48 | 45.76 | 36.67 | 45.49 | 54.89 | 48.67 |
| | 1shot-Text-S0.3 | 30.18 | 31.05 | 29.77 | 36.14 | 37.59 | 36.68 |
| | 1shot-Text-S0.3-COT | 37.44 | 36.65 | 35.74 | 42.01 | 42.32 | 42.14 |
| | 1shot-Vision | 38.98 | 42.27 | 37.93 | 47.54 | 50.00 | 48.43 |
| | 1shot-Vision-Calc | 33.76 | 39.43 | 32.67 | 40.33 | 44.13 | 41.61 |
| | 1shot-Vision-COT | 44.90 | 44.90 | 44.29 | 46.92 | 47.32 | 47.08 |
| | 1shot-Vision-Dyscalc | 39.61 | 43.23 | 38.34 | 48.48 | 51.02 | 49.37 |
| Isolation-Forest | 0shot | 6.04 | 19.74 | 8.64 | 21.81 | 42.14 | 28.69 |
| Qwen | 0shot-Text | 0.00 | 0.00 | 0.00 | 0.00 | 0.00 | 0.00 |
| | 0shot-Text-S0.3 | 0.00 | 0.00 | 0.00 | 0.00 | 0.00 | 0.00 |
| | 0shot-Text-S0.3-Calc | 0.00 | 0.00 | 0.00 | 0.00 | 0.00 | 0.00 |
| | 0shot-Text-S0.3-COT | 3.25 | 3.25 | 3.25 | 3.49 | 3.72 | 3.56 |
| | 0shot-Text-S0.3-COT-CSV | 0.00 | 0.00 | 0.00 | 0.00 | 0.00 | 0.00 |
| | 0shot-Text-S0.3-COT-PAP | 6.26 | 6.50 | 6.27 | 6.41 | 6.56 | 6.46 |
| | 0shot-Text-S0.3-COT-TPD | 0.75 | 0.75 | 0.75 | 0.80 | 0.85 | 0.82 |
| | 0shot-Text-S0.3-CSV | 0.00 | 0.00 | 0.00 | 0.00 | 0.00 | 0.00 |
| | 0shot-Text-S0.3-Dyscalc | 0.00 | 0.00 | 0.00 | 0.00 | 0.00 | 0.00 |
| | 0shot-Text-S0.3-PAP | 12.75 | 12.75 | 12.75 | 12.75 | 12.76 | 12.76 |
| | 0shot-Text-S0.3-TPD | 1.25 | 1.25 | 1.25 | 1.30 | 1.36 | 1.32 |
| | 0shot-Vision | 8.21 | 32.50 | 10.13 | 21.04 | 34.49 | 25.66 |
| | 0shot-Vision-Calc | 5.68 | 27.76 | 7.52 | 16.91 | 29.03 | 21.07 |
| | 0shot-Vision-COT | 12.25 | 17.59 | 12.66 | 15.80 | 19.06 | 16.94 |
| | 0shot-Vision-Dyscalc | 6.51 | 25.75 | 8.10 | 15.71 | 25.77 | 19.16 |
| | 1shot-Text-S0.3-COT | 0.00 | 0.00 | 0.00 | 0.00 | 0.00 | 0.00 |
| | 1shot-Vision | 7.80 | 28.89 | 9.46 | 25.09 | 37.34 | 29.43 |
| | 1shot-Vision-Calc | 13.13 | 20.00 | 13.62 | 16.54 | 20.44 | 17.86 |
| | 1shot-Vision-COT | 27.57 | 28.08 | 27.56 | 29.62 | 30.16 | 29.85 |
| | 1shot-Vision-Dyscalc | 14.85 | 31.64 | 15.48 | 29.00 | 38.45 | 32.33 |
| Threshold | 0shot | 6.91 | 2.98 | 3.90 | 20.15 | 38.90 | 26.45 |

Table 3: Frequency anomalies in regular sine wave

| | | PRE | REC | F1 | affi PRE | affi REC | affi F1 |
|---|---|---|---|---|---|---|---|
| Gemini-1.5-Flash | 0shot-Text | 4.01 | 4.30 | 3.63 | 39.24 | 31.84 | 33.65 |
| | 0shot-Text-S0.3 | 11.28 | 14.24 | 10.91 | 58.42 | 52.67 | 53.59 |
| | 0shot-Text-S0.3-Calc | 3.29 | 2.61 | 2.51 | 36.76 | 27.58 | 29.91 |
| | 0shot-Text-S0.3-COT | 3.46 | 4.78 | 3.36 | 21.76 | 19.93 | 19.99 |
| | 0shot-Text-S0.3-COT-CSV | 2.55 | 2.34 | 1.94 | 30.81 | 23.37 | 25.37 |
| | 0shot-Text-S0.3-COT-PAP | 3.92 | 2.35 | 2.61 | 29.46 | 19.21 | 22.12 |
| | 0shot-Text-S0.3-COT-TPD | 2.61 | 2.37 | 1.97 | 25.96 | 19.14 | 21.00 |
| | 0shot-Text-S0.3-CSV | 2.77 | 2.21 | 2.08 | 46.48 | 33.66 | 37.03 |
| | 0shot-Text-S0.3-Dyscalc | 3.74 | 3.24 | 2.91 | 36.95 | 28.19 | 30.32 |
| | 0shot-Text-S0.3-PAP | 1.97 | 0.73 | 0.94 | 33.36 | 22.71 | 25.54 |
| | 0shot-Text-S0.3-TPD | 2.57 | 1.57 | 1.63 | 39.54 | 29.61 | 32.22 |

Table 3 – continued from previous page

| | | PRE | REC | F1 | affi PRE | affi REC | affi F1 |
|---|---|---|---|---|---|---|---|
| | 0shot-Vision | 13.56 | 13.77 | 13.54 | 15.47 | 13.15 | 13.76 |
| | 0shot-Vision-Calc | 17.82 | 19.68 | 18.14 | 23.65 | 17.80 | 19.39 |
| | 0shot-Vision-COT | 12.65 | 12.71 | 12.45 | 14.16 | 12.52 | 12.98 |
| | 0shot-Vision-Dyscalc | 16.71 | 17.84 | 16.83 | 21.44 | 16.58 | 17.88 |
| | 1shot-Text-S0.3 | 10.20 | 14.73 | 10.45 | 56.37 | 52.78 | 52.61 |
| | 1shot-Text-S0.3-COT | 12.81 | 17.30 | 12.31 | 56.67 | 50.63 | 51.30 |
| | 1shot-Vision | 21.78 | 21.31 | 20.52 | 35.84 | 25.89 | 28.46 |
| | 1shot-Vision-Calc | 23.10 | 20.50 | 20.73 | 34.76 | 24.87 | 27.46 |
| | 1shot-Vision-COT | 14.29 | 16.89 | 13.80 | 54.19 | 41.68 | 43.93 |
| | 1shot-Vision-Dyscalc | 23.73 | 21.83 | 21.61 | 36.35 | 24.37 | 27.57 |
| GPT-4o-Mini | 0shot-Text | 11.14 | 10.91 | 10.71 | 23.10 | 20.56 | 21.21 |
| | 0shot-Text-S0.3 | 9.18 | 10.37 | 8.75 | 53.75 | 49.13 | 49.18 |
| | 0shot-Text-S0.3-Calc | 3.51 | 3.11 | 2.83 | 33.03 | 28.20 | 29.02 |
| | 0shot-Text-S0.3-COT | 3.95 | 8.88 | 4.53 | 33.36 | 36.97 | 33.79 |
| | 0shot-Text-S0.3-COT-CSV | 4.56 | 4.73 | 4.36 | 14.51 | 11.35 | 12.07 |
| | 0shot-Text-S0.3-COT-PAP | 6.13 | 6.30 | 6.12 | 10.72 | 9.50 | 9.82 |
| | 0shot-Text-S0.3-COT-TPD | 0.32 | 0.31 | 0.29 | 4.23 | 3.05 | 3.37 |
| | 0shot-Text-S0.3-CSV | 10.52 | 9.74 | 9.80 | 24.45 | 19.26 | 20.67 |
| | 0shot-Text-S0.3-Dyscalc | 2.95 | 2.64 | 2.24 | 36.62 | 30.97 | 31.90 |
| | 0shot-Text-S0.3-PAP | 8.09 | 7.95 | 7.68 | 19.09 | 14.66 | 15.84 |
| | 0shot-Text-S0.3-TPD | 2.66 | 2.75 | 2.16 | 31.51 | 26.09 | 27.19 |
| | 0shot-Vision | 10.00 | 10.00 | 10.00 | 10.00 | 10.00 | 10.00 |
| | 0shot-Vision-Calc | 10.38 | 10.16 | 10.21 | 10.89 | 10.70 | 10.76 |
| | 0shot-Vision-COT | 10.25 | 10.05 | 10.08 | 10.25 | 10.12 | 10.16 |
| | 0shot-Vision-Dyscalc | 10.45 | 10.18 | 10.25 | 11.10 | 10.82 | 10.91 |
| | 1shot-Text-S0.3 | 9.58 | 11.05 | 9.08 | 52.93 | 48.83 | 48.65 |
| | 1shot-Text-S0.3-COT | 6.91 | 13.05 | 7.70 | 51.68 | 54.35 | 51.29 |
| | 1shot-Vision | 12.42 | 10.32 | 10.69 | 34.12 | 26.85 | 28.63 |
| | 1shot-Vision-Calc | 11.91 | 11.21 | 11.06 | 24.28 | 19.32 | 20.70 |
| | 1shot-Vision-COT | 11.86 | 11.04 | 11.09 | 19.80 | 17.07 | 17.67 |
| | 1shot-Vision-Dyscalc | 12.32 | 11.03 | 11.08 | 25.08 | 19.94 | 21.15 |
| Internvlm-76B | 0shot-Text | 5.66 | 12.39 | 6.04 | 15.82 | 17.30 | 15.53 |
| | 0shot-Text-S0.3 | 3.38 | 7.18 | 3.33 | 37.12 | 33.84 | 33.16 |
| | 0shot-Text-S0.3-Calc | 8.18 | 9.48 | 8.30 | 14.36 | 13.50 | 13.45 |
| | 0shot-Text-S0.3-COT | 6.51 | 7.71 | 6.72 | 12.89 | 11.98 | 12.06 |
| | 0shot-Text-S0.3-COT-CSV | 2.60 | 4.52 | 2.69 | 19.44 | 17.81 | 17.71 |
| | 0shot-Text-S0.3-COT-PAP | 3.79 | 3.68 | 3.71 | 6.45 | 5.57 | 5.78 |
| | 0shot-Text-S0.3-COT-TPD | 3.13 | 3.73 | 3.15 | 10.65 | 8.46 | 8.98 |
| | 0shot-Text-S0.3-CSV | 3.65 | 8.97 | 3.62 | 37.52 | 31.89 | 32.68 |
| | 0shot-Text-S0.3-Dyscalc | 7.15 | 8.20 | 7.22 | 16.55 | 13.84 | 14.39 |
| | 0shot-Text-S0.3-PAP | 8.36 | 9.29 | 8.38 | 13.77 | 12.68 | 12.77 |
| | 0shot-Text-S0.3-TPD | 4.01 | 7.60 | 4.01 | 24.88 | 21.46 | 21.45 |
| | 0shot-Vision | 6.51 | 17.82 | 8.81 | 35.94 | 35.14 | 33.37 |
| | 0shot-Vision-Calc | 4.68 | 15.55 | 5.41 | 26.42 | 29.19 | 25.80 |
| | 0shot-Vision-COT | 4.97 | 9.25 | 5.37 | 30.18 | 27.70 | 27.27 |
| | 0shot-Vision-Dyscalc | 7.16 | 18.74 | 8.88 | 32.78 | 33.72 | 31.01 |
| | 1shot-Text-S0.3 | 4.24 | 4.93 | 3.22 | 37.50 | 32.79 | 32.66 |
| | 1shot-Text-S0.3-COT | 3.05 | 2.69 | 2.40 | 30.54 | 26.81 | 26.86 |
| | 1shot-Vision | 4.31 | 15.22 | 4.92 | 38.93 | 44.67 | 38.43 |
| | 1shot-Vision-Calc | 4.55 | 13.45 | 5.19 | 35.04 | 38.20 | 33.78 |
| | 1shot-Vision-COT | 2.76 | 8.41 | 3.58 | 35.98 | 42.04 | 36.75 |
| | 1shot-Vision-Dyscalc | 4.11 | 12.28 | 5.37 | 35.10 | 38.41 | 33.73 |
| Isolation-Forest | 0shot | 3.41 | 90.00 | 6.44 | 45.27 | 90.00 | 60.25 |

Table 3 – continued from previous page

|  |  | PRE | REC | F1 | affi PRE | affi REC | affi F1 |
|---|---|---|---|---|---|---|---|
| Qwen | 0shot-Text | 0.00 | 0.00 | 0.00 | 0.00 | 0.00 | 0.00 |
|  | 0shot-Text-S0.3 | 0.00 | 0.00 | 0.00 | 0.00 | 0.00 | 0.00 |
|  | 0shot-Text-S0.3-Calc | 0.00 | 0.00 | 0.00 | 0.00 | 0.00 | 0.00 |
|  | 0shot-Text-S0.3-COT | 0.55 | 0.92 | 0.59 | 1.52 | 1.36 | 1.37 |
|  | 0shot-Text-S0.3-COT-CSV | 0.00 | 0.00 | 0.00 | 0.00 | 0.00 | 0.00 |
|  | 0shot-Text-S0.3-COT-PAP | 1.28 | 1.50 | 1.31 | 1.73 | 1.64 | 1.65 |
|  | 0shot-Text-S0.3-COT-TPD | 0.54 | 0.61 | 0.53 | 1.68 | 1.11 | 1.27 |
|  | 0shot-Text-S0.3-CSV | 0.00 | 0.00 | 0.00 | 0.00 | 0.00 | 0.00 |
|  | 0shot-Text-S0.3-Dyscalc | 0.00 | 0.00 | 0.00 | 0.00 | 0.00 | 0.00 |
|  | 0shot-Text-S0.3-PAP | 2.25 | 2.25 | 2.25 | 3.77 | 3.13 | 3.30 |
|  | 0shot-Text-S0.3-TPD | 0.07 | 0.81 | 0.12 | 3.14 | 2.67 | 2.73 |
|  | 0shot-Vision | 3.17 | 55.44 | 5.25 | 32.89 | 60.12 | 42.26 |
|  | 0shot-Vision-Calc | 2.79 | 56.91 | 4.81 | 30.73 | 58.87 | 40.24 |
|  | 0shot-Vision-COT | 4.44 | 8.61 | 4.60 | 7.13 | 9.26 | 7.86 |
|  | 0shot-Vision-Dyscalc | 3.19 | 42.57 | 4.68 | 24.03 | 44.83 | 31.05 |
|  | 1shot-Text-S0.3-COT | 0.00 | 0.00 | 0.00 | 0.00 | 0.00 | 0.00 |
|  | 1shot-Vision | 4.31 | 17.42 | 4.66 | 43.41 | 50.74 | 44.11 |
|  | 1shot-Vision-Calc | 1.12 | 3.31 | 1.23 | 2.97 | 4.11 | 3.32 |
|  | 1shot-Vision-COT | 1.76 | 3.56 | 1.88 | 11.06 | 14.18 | 12.02 |
|  | 1shot-Vision-Dyscalc | 5.06 | 13.32 | 5.11 | 39.87 | 44.46 | 39.41 |
| Threshold | 0shot | 3.20 | 3.47 | 2.95 | 44.73 | 85.12 | 58.59 |

Table 4: Point noises anomalies in regular sine wave

|  |  | PRE | REC | F1 | affi PRE | affi REC | affi F1 |
|---|---|---|---|---|---|---|---|
| Gemini-1.5-Flash | 0shot-Text | 1.09 | 1.06 | 0.93 | 23.84 | 24.38 | 23.35 |
|  | 0shot-Text-S0.3 | 2.75 | 3.60 | 2.83 | 43.01 | 41.02 | 40.64 |
|  | 0shot-Text-S0.3-Calc | 0.10 | 0.05 | 0.06 | 18.67 | 19.37 | 18.53 |
|  | 0shot-Text-S0.3-COT | 2.66 | 3.72 | 2.54 | 19.37 | 20.22 | 19.12 |
|  | 0shot-Text-S0.3-COT-CSV | 2.88 | 2.86 | 2.19 | 24.03 | 20.64 | 21.46 |
|  | 0shot-Text-S0.3-COT-PAP | 3.31 | 3.51 | 3.30 | 17.68 | 16.21 | 16.44 |
|  | 0shot-Text-S0.3-COT-TPD | 3.89 | 3.79 | 3.76 | 18.41 | 16.85 | 17.14 |
|  | 0shot-Text-S0.3-CSV | 3.81 | 2.85 | 2.74 | 34.27 | 29.22 | 30.44 |
|  | 0shot-Text-S0.3-Dyscalc | 0.20 | 0.10 | 0.12 | 18.22 | 18.78 | 18.08 |
|  | 0shot-Text-S0.3-PAP | 0.00 | 0.00 | 0.00 | 15.58 | 14.10 | 14.28 |
|  | 0shot-Text-S0.3-TPD | 1.83 | 1.76 | 1.77 | 22.66 | 20.71 | 20.94 |
|  | 0shot-Vision | 52.35 | 90.69 | 60.90 | 94.22 | 95.46 | 94.38 |
|  | 0shot-Vision-Calc | 51.56 | 87.56 | 59.74 | 94.28 | 95.88 | 94.63 |
|  | 0shot-Vision-COT | 51.87 | 84.03 | 59.20 | 91.78 | 89.43 | 89.75 |
|  | 0shot-Vision-Dyscalc | 51.40 | 88.72 | 59.53 | 94.18 | 95.87 | 94.52 |
|  | 1shot-Text-S0.3 | 3.63 | 4.20 | 3.61 | 43.49 | 41.13 | 40.93 |
|  | 1shot-Text-S0.3-COT | 8.19 | 9.11 | 8.05 | 45.60 | 43.79 | 43.26 |
|  | 1shot-Vision | 53.54 | 78.76 | 59.96 | 93.42 | 93.83 | 93.19 |
|  | 1shot-Vision-Calc | 55.95 | 77.36 | 61.54 | 94.88 | 94.55 | 94.17 |
|  | 1shot-Vision-COT | 43.09 | 67.37 | 49.31 | 82.62 | 83.34 | 82.61 |
|  | 1shot-Vision-Dyscalc | 54.29 | 78.90 | 60.44 | 95.18 | 96.15 | 95.25 |
| GPT-4o-Mini | 0shot-Text | 30.54 | 30.52 | 30.32 | 43.21 | 40.21 | 40.97 |
|  | 0shot-Text-S0.3 | 9.01 | 8.50 | 7.20 | 43.15 | 41.55 | 40.79 |
|  | 0shot-Text-S0.3-Calc | 1.98 | 2.21 | 1.94 | 21.50 | 19.72 | 19.99 |
|  | 0shot-Text-S0.3-COT | 6.13 | 8.93 | 5.87 | 35.72 | 40.27 | 36.77 |

Table 4 – continued from previous page

|  |  | PRE | REC | F1 | affi PRE | affi REC | affi F1 |
|---|---|---|---|---|---|---|---|
|  | 0shot-Text-S0.3-COT-CSV | 7.34 | 7.42 | 6.84 | 27.29 | 22.92 | 24.05 |
|  | 0shot-Text-S0.3-COT-PAP | 12.50 | 12.50 | 12.50 | 16.22 | 15.71 | 15.83 |
|  | 0shot-Text-S0.3-COT-TPD | 0.93 | 0.86 | 0.87 | 8.27 | 7.94 | 7.89 |
|  | 0shot-Text-S0.3-CSV | 16.60 | 17.03 | 16.34 | 30.86 | 29.03 | 29.29 |
|  | 0shot-Text-S0.3-Dyscalc | 1.49 | 1.89 | 1.51 | 24.42 | 23.94 | 23.45 |
|  | 0shot-Text-S0.3-PAP | 15.02 | 14.88 | 14.93 | 23.92 | 22.86 | 23.05 |
|  | 0shot-Text-S0.3-TPD | 4.11 | 6.40 | 4.34 | 22.28 | 23.67 | 22.35 |
|  | 0shot-Vision | 39.16 | 41.75 | 39.01 | 62.88 | 58.88 | 60.10 |
|  | 0shot-Vision-Calc | 38.89 | 42.58 | 39.11 | 63.17 | 61.04 | 61.42 |
|  | 0shot-Vision-COT | 39.70 | 40.55 | 38.90 | 63.08 | 58.39 | 59.89 |
|  | 0shot-Vision-Dyscalc | 39.02 | 42.07 | 39.13 | 63.05 | 60.52 | 61.12 |
|  | 1shot-Text-S0.3 | 6.11 | 4.87 | 4.26 | 42.74 | 40.82 | 40.18 |
|  | 1shot-Text-S0.3-COT | 10.90 | 14.44 | 11.04 | 46.00 | 50.70 | 47.15 |
|  | 1shot-Vision | 38.24 | 39.36 | 36.32 | 76.21 | 71.14 | 72.22 |
|  | 1shot-Vision-Calc | 41.29 | 41.60 | 39.22 | 76.55 | 71.06 | 72.36 |
|  | 1shot-Vision-COT | 40.35 | 42.84 | 39.01 | 76.07 | 71.23 | 72.04 |
|  | 1shot-Vision-Dyscalc | 40.96 | 43.16 | 39.99 | 77.02 | 71.87 | 72.99 |
| Internvlm-76B | 0shot-Text | 19.26 | 22.48 | 19.47 | 25.08 | 27.32 | 25.84 |
|  | 0shot-Text-S0.3 | 10.21 | 11.34 | 9.96 | 33.19 | 34.47 | 32.60 |
|  | 0shot-Text-S0.3-Calc | 21.42 | 23.25 | 21.52 | 25.62 | 26.72 | 25.93 |
|  | 0shot-Text-S0.3-COT | 15.82 | 16.50 | 15.86 | 18.51 | 19.07 | 18.68 |
|  | 0shot-Text-S0.3-COT-CSV | 9.21 | 9.82 | 8.81 | 26.12 | 24.75 | 24.70 |
|  | 0shot-Text-S0.3-COT-PAP | 10.58 | 10.83 | 10.58 | 11.56 | 11.76 | 11.62 |
|  | 0shot-Text-S0.3-COT-TPD | 9.00 | 9.00 | 9.00 | 11.69 | 11.68 | 11.63 |
|  | 0shot-Text-S0.3-CSV | 11.92 | 13.91 | 11.20 | 40.05 | 38.01 | 37.78 |
|  | 0shot-Text-S0.3-Dyscalc | 18.89 | 19.66 | 18.94 | 23.35 | 23.80 | 23.46 |
|  | 0shot-Text-S0.3-PAP | 25.26 | 25.50 | 25.27 | 27.14 | 27.26 | 27.16 |
|  | 0shot-Text-S0.3-TPD | 7.00 | 7.00 | 7.00 | 14.89 | 15.08 | 14.88 |
|  | 0shot-Vision | 14.26 | 53.10 | 20.64 | 56.18 | 67.83 | 60.77 |
|  | 0shot-Vision-Calc | 22.97 | 58.75 | 28.77 | 66.27 | 76.14 | 70.44 |
|  | 0shot-Vision-COT | 4.35 | 12.95 | 5.47 | 31.51 | 39.00 | 33.61 |
|  | 0shot-Vision-Dyscalc | 17.31 | 54.36 | 23.36 | 59.49 | 70.15 | 63.79 |
|  | 1shot-Text-S0.3 | 8.69 | 9.89 | 7.70 | 32.37 | 32.95 | 31.19 |
|  | 1shot-Text-S0.3-COT | 10.20 | 9.98 | 9.43 | 30.63 | 30.69 | 29.49 |
|  | 1shot-Vision | 9.97 | 29.84 | 12.99 | 44.76 | 57.62 | 49.26 |
|  | 1shot-Vision-Calc | 10.01 | 29.92 | 12.26 | 44.34 | 57.93 | 48.86 |
|  | 1shot-Vision-COT | 3.87 | 12.65 | 4.61 | 33.66 | 45.65 | 37.30 |
|  | 1shot-Vision-Dyscalc | 8.98 | 31.81 | 11.90 | 42.77 | 59.47 | 48.65 |
| Isolation-Forest | 0shot | 3.36 | 70.48 | 6.16 | 35.66 | 70.75 | 47.43 |
| Qwen | 0shot-Text | 0.00 | 0.00 | 0.00 | 0.00 | 0.00 | 0.00 |
|  | 0shot-Text-S0.3 | 0.00 | 0.00 | 0.00 | 0.00 | 0.00 | 0.00 |
|  | 0shot-Text-S0.3-Calc | 0.00 | 0.00 | 0.00 | 0.00 | 0.00 | 0.00 |
|  | 0shot-Text-S0.3-COT | 1.50 | 1.50 | 1.50 | 1.61 | 1.59 | 1.60 |
|  | 0shot-Text-S0.3-COT-CSV | 0.00 | 0.00 | 0.00 | 0.00 | 0.00 | 0.00 |
|  | 0shot-Text-S0.3-COT-PAP | 2.00 | 2.00 | 2.00 | 2.00 | 2.00 | 2.00 |
|  | 0shot-Text-S0.3-COT-TPD | 0.50 | 0.50 | 0.50 | 0.80 | 0.84 | 0.82 |
|  | 0shot-Text-S0.3-CSV | 0.00 | 0.00 | 0.00 | 0.00 | 0.00 | 0.00 |
|  | 0shot-Text-S0.3-Dyscalc | 0.00 | 0.00 | 0.00 | 0.00 | 0.00 | 0.00 |
|  | 0shot-Text-S0.3-PAP | 6.00 | 6.00 | 6.00 | 6.16 | 6.17 | 6.16 |
|  | 0shot-Text-S0.3-TPD | 0.50 | 0.50 | 0.50 | 1.19 | 1.34 | 1.25 |
|  | 0shot-Vision | 6.29 | 42.00 | 8.22 | 30.05 | 48.49 | 36.66 |
|  | 0shot-Vision-Calc | 4.11 | 36.69 | 5.56 | 22.05 | 39.14 | 27.93 |
|  | 0shot-Vision-COT | 12.70 | 19.27 | 13.14 | 17.54 | 21.53 | 18.92 |

Table 4 – continued from previous page

|  |  | PRE | REC | F1 | affi PRE | affi REC | affi F1 |
|---|---|---|---|---|---|---|---|
|  | 0shot-Vision-Dyscalc | 3.26 | 28.28 | 4.58 | 17.87 | 30.85 | 22.36 |
|  | 1shot-Text-S0.3-COT | 0.00 | 0.00 | 0.00 | 0.00 | 0.00 | 0.00 |
|  | 1shot-Vision | 9.15 | 25.49 | 10.04 | 41.59 | 47.97 | 42.61 |
|  | 1shot-Vision-Calc | 7.88 | 12.80 | 8.16 | 10.78 | 13.28 | 11.59 |
|  | 1shot-Vision-COT | 11.42 | 16.01 | 11.60 | 29.38 | 30.82 | 29.23 |
|  | 1shot-Vision-Dyscalc | 8.55 | 19.61 | 8.72 | 37.64 | 43.60 | 38.63 |
| Threshold | 0shot | 3.49 | 3.17 | 2.82 | 35.06 | 68.55 | 46.36 |

Table 5: Out-of-range anomalies in Gaussian noise

|  |  | PRE | REC | F1 | affi PRE | affi REC | affi F1 |
|---|---|---|---|---|---|---|---|
| Gemini-1.5-Flash | 0shot-Text | 2.92 | 4.53 | 2.98 | 32.52 | 38.30 | 34.27 |
|  | 0shot-Text-S0.3 | 7.15 | 12.76 | 8.34 | 46.77 | 56.31 | 50.25 |
|  | 0shot-Text-S0.3-Calc | 2.24 | 1.92 | 1.81 | 35.13 | 36.94 | 34.99 |
|  | 0shot-Text-S0.3-COT | 4.28 | 4.90 | 4.18 | 20.54 | 22.27 | 20.84 |
|  | 0shot-Text-S0.3-COT-CSV | 2.70 | 2.24 | 2.34 | 16.85 | 15.47 | 15.75 |
|  | 0shot-Text-S0.3-COT-PAP | 6.59 | 6.17 | 6.17 | 25.46 | 22.12 | 23.01 |
|  | 0shot-Text-S0.3-COT-TPD | 2.87 | 3.03 | 2.86 | 16.75 | 14.98 | 15.26 |
|  | 0shot-Text-S0.3-CSV | 5.00 | 4.52 | 4.62 | 35.17 | 33.31 | 33.28 |
|  | 0shot-Text-S0.3-Dyscalc | 2.20 | 1.95 | 1.85 | 33.94 | 36.51 | 34.22 |
|  | 0shot-Text-S0.3-PAP | 1.97 | 0.88 | 1.08 | 30.66 | 27.80 | 28.09 |
|  | 0shot-Text-S0.3-TPD | 2.30 | 2.63 | 2.13 | 29.22 | 28.56 | 27.96 |
|  | 0shot-Vision | 32.70 | 74.46 | 40.19 | 79.35 | 83.16 | 80.95 |
|  | 0shot-Vision-Calc | 20.96 | 60.42 | 28.02 | 68.04 | 70.94 | 69.07 |
|  | 0shot-Vision-COT | 28.58 | 61.32 | 34.67 | 75.91 | 73.62 | 73.75 |
|  | 0shot-Vision-Dyscalc | 26.94 | 66.70 | 34.00 | 74.17 | 77.29 | 75.34 |
|  | 1shot-Text-S0.3 | 7.12 | 13.71 | 8.42 | 47.06 | 56.95 | 50.69 |
|  | 1shot-Text-S0.3-COT | 12.04 | 14.57 | 12.31 | 46.38 | 49.43 | 46.57 |
|  | 1shot-Vision | 33.77 | 63.17 | 40.02 | 80.05 | 82.67 | 81.07 |
|  | 1shot-Vision-Calc | 36.28 | 64.25 | 42.40 | 81.41 | 84.05 | 82.40 |
|  | 1shot-Vision-COT | 22.85 | 50.51 | 28.87 | 69.16 | 71.62 | 70.05 |
|  | 1shot-Vision-Dyscalc | 35.52 | 61.50 | 41.15 | 81.49 | 83.06 | 81.82 |
| GPT-4o-Mini | 0shot-Text | 8.08 | 9.57 | 8.19 | 38.07 | 36.57 | 36.13 |
|  | 0shot-Text-S0.3 | 7.45 | 11.75 | 8.13 | 42.93 | 45.39 | 42.84 |
|  | 0shot-Text-S0.3-Calc | 17.02 | 16.80 | 16.80 | 30.47 | 28.11 | 28.71 |
|  | 0shot-Text-S0.3-COT | 10.00 | 10.61 | 9.21 | 31.39 | 30.25 | 30.00 |
|  | 0shot-Text-S0.3-COT-CSV | 10.40 | 10.05 | 9.98 | 24.63 | 21.52 | 22.37 |
|  | 0shot-Text-S0.3-COT-PAP | 6.74 | 6.87 | 6.72 | 15.02 | 12.78 | 13.43 |
|  | 0shot-Text-S0.3-COT-TPD | 0.84 | 0.80 | 0.82 | 4.45 | 4.50 | 4.41 |
|  | 0shot-Text-S0.3-CSV | 20.31 | 19.38 | 19.23 | 42.18 | 37.17 | 38.57 |
|  | 0shot-Text-S0.3-Dyscalc | 12.86 | 12.95 | 12.62 | 26.48 | 24.86 | 24.96 |
|  | 0shot-Text-S0.3-PAP | 15.99 | 15.63 | 15.66 | 26.82 | 23.95 | 24.75 |
|  | 0shot-Text-S0.3-TPD | 8.11 | 7.29 | 7.41 | 24.51 | 23.05 | 23.04 |
|  | 0shot-Vision | 37.13 | 44.84 | 38.48 | 74.12 | 77.89 | 75.28 |
|  | 0shot-Vision-Calc | 37.30 | 43.03 | 38.00 | 75.49 | 78.54 | 76.26 |
|  | 0shot-Vision-COT | 36.58 | 44.09 | 37.55 | 75.99 | 78.92 | 76.74 |
|  | 0shot-Vision-Dyscalc | 36.67 | 41.84 | 37.21 | 72.87 | 75.26 | 73.43 |
|  | 1shot-Text-S0.3 | 7.96 | 12.51 | 8.52 | 43.20 | 45.55 | 43.17 |
|  | 1shot-Text-S0.3-COT | 9.43 | 11.72 | 9.65 | 44.53 | 43.69 | 42.97 |
|  | 1shot-Vision | 31.52 | 33.83 | 30.99 | 75.81 | 76.46 | 75.34 |

Table 5 – continued from previous page

| | | PRE | REC | F1 | affi PRE | affi REC | affi F1 |
|---|---|---|---|---|---|---|---|
| | 1shot-Vision-Calc | 32.83 | 36.30 | 32.24 | 72.08 | 72.94 | 71.72 |
| | 1shot-Vision-COT | 30.29 | 34.15 | 30.50 | 76.48 | 77.37 | 76.14 |
| | 1shot-Vision-Dyscalc | 31.18 | 32.85 | 30.59 | 73.85 | 74.29 | 73.26 |
| Internvlm-76B | 0shot-Text | 8.95 | 27.30 | 9.39 | 27.16 | 36.32 | 29.81 |
| | 0shot-Text-S0.3 | 7.45 | 8.89 | 7.48 | 25.94 | 25.18 | 24.59 |
| | 0shot-Text-S0.3-Calc | 10.49 | 13.55 | 10.29 | 20.98 | 21.34 | 20.46 |
| | 0shot-Text-S0.3-COT | 9.85 | 10.45 | 9.94 | 16.89 | 17.87 | 17.13 |
| | 0shot-Text-S0.3-COT-CSV | 4.83 | 5.71 | 4.75 | 19.71 | 18.08 | 18.24 |
| | 0shot-Text-S0.3-COT-PAP | 12.58 | 12.40 | 12.37 | 18.20 | 17.09 | 17.31 |
| | 0shot-Text-S0.3-COT-TPD | 6.40 | 6.46 | 6.34 | 11.74 | 11.09 | 11.12 |
| | 0shot-Text-S0.3-CSV | 7.83 | 8.24 | 7.49 | 38.79 | 34.59 | 35.52 |
| | 0shot-Text-S0.3-Dyscalc | 7.56 | 16.99 | 7.92 | 21.17 | 25.37 | 22.00 |
| | 0shot-Text-S0.3-PAP | 25.43 | 27.34 | 25.51 | 28.87 | 29.71 | 29.00 |
| | 0shot-Text-S0.3-TPD | 7.83 | 12.72 | 8.05 | 22.23 | 23.44 | 21.88 |
| | 0shot-Vision | 27.08 | 56.67 | 32.58 | 73.52 | 79.03 | 75.84 |
| | 0shot-Vision-Calc | 27.58 | 56.64 | 32.82 | 72.83 | 78.02 | 74.93 |
| | 0shot-Vision-COT | 8.25 | 12.05 | 8.64 | 34.21 | 36.28 | 33.99 |
| | 0shot-Vision-Dyscalc | 4.39 | 13.78 | 5.03 | 32.84 | 39.45 | 34.03 |
| | 1shot-Text-S0.3 | 8.61 | 9.95 | 8.37 | 27.64 | 26.60 | 25.94 |
| | 1shot-Text-S0.3-COT | 8.70 | 8.90 | 8.48 | 28.66 | 28.08 | 27.47 |
| | 1shot-Vision | 17.79 | 33.39 | 20.35 | 58.16 | 61.04 | 58.53 |
| | 1shot-Vision-Calc | 19.32 | 35.23 | 21.91 | 60.17 | 65.54 | 61.65 |
| | 1shot-Vision-COT | 3.57 | 5.90 | 3.66 | 31.10 | 34.97 | 31.58 |
| | 1shot-Vision-Dyscalc | 17.24 | 34.25 | 20.14 | 59.85 | 65.48 | 61.53 |
| Isolation-Forest | 0shot | 15.77 | 69.75 | 23.58 | 42.30 | 69.75 | 52.44 |
| Qwen | 0shot-Text | 0.00 | 0.00 | 0.00 | 0.00 | 0.00 | 0.00 |
| | 0shot-Text-S0.3 | 0.00 | 0.00 | 0.00 | 0.00 | 0.00 | 0.00 |
| | 0shot-Text-S0.3-Calc | 0.00 | 0.00 | 0.00 | 0.00 | 0.00 | 0.00 |
| | 0shot-Text-S0.3-COT | 1.26 | 1.51 | 1.27 | 2.13 | 2.06 | 2.05 |
| | 0shot-Text-S0.3-COT-CSV | 0.00 | 0.00 | 0.00 | 0.00 | 0.00 | 0.00 |
| | 0shot-Text-S0.3-COT-PAP | 4.50 | 4.50 | 4.50 | 4.64 | 4.64 | 4.64 |
| | 0shot-Text-S0.3-COT-TPD | 0.61 | 0.85 | 0.57 | 1.59 | 1.51 | 1.44 |
| | 0shot-Text-S0.3-CSV | 0.00 | 0.00 | 0.00 | 0.00 | 0.00 | 0.00 |
| | 0shot-Text-S0.3-Dyscalc | 0.00 | 0.00 | 0.00 | 0.00 | 0.00 | 0.00 |
| | 0shot-Text-S0.3-PAP | 7.25 | 7.25 | 7.25 | 7.25 | 7.25 | 7.25 |
| | 0shot-Text-S0.3-TPD | 1.21 | 1.08 | 1.12 | 1.72 | 1.66 | 1.65 |
| | 0shot-Vision | 3.22 | 29.46 | 4.75 | 30.09 | 47.51 | 36.40 |
| | 0shot-Vision-Calc | 2.71 | 35.65 | 4.60 | 31.95 | 52.36 | 39.18 |
| | 0shot-Vision-COT | 9.63 | 16.43 | 9.92 | 13.94 | 17.67 | 15.23 |
| | 0shot-Vision-Dyscalc | 2.90 | 28.12 | 4.34 | 26.32 | 41.84 | 31.78 |
| | 1shot-Text-S0.3-COT | 0.00 | 0.00 | 0.00 | 0.00 | 0.00 | 0.00 |
| | 1shot-Vision | 9.42 | 17.36 | 10.10 | 38.10 | 43.64 | 39.14 |
| | 1shot-Vision-Calc | 7.13 | 9.19 | 7.21 | 9.97 | 10.89 | 10.23 |
| | 1shot-Vision-COT | 11.12 | 12.31 | 10.95 | 22.28 | 24.65 | 22.70 |
| | 1shot-Vision-Dyscalc | 8.92 | 13.66 | 9.20 | 34.59 | 37.55 | 34.54 |
| Threshold | 0shot | 32.49 | 53.38 | 34.28 | 50.54 | 69.54 | 58.09 |

Table 6: Trend anomalies in shifting sine wave with extra noise

| | | PRE | REC | F1 | affi PRE | affi REC | affi F1 |
|---|---|---|---|---|---|---|---|
| Gemini-1.5-Flash | 0shot-Text | 0.00 | 0.00 | 0.00 | 7.07 | 13.05 | 9.16 |
| | 0shot-Text-S0.3 | 0.43 | 0.62 | 0.46 | 12.77 | 24.24 | 16.68 |
| | 0shot-Text-S0.3-COT | 1.25 | 2.43 | 1.51 | 8.95 | 15.05 | 11.15 |
| | 0shot-Text-S0.3-COT-CSV | 0.00 | 0.00 | 0.00 | 3.50 | 6.32 | 4.43 |
| | 0shot-Text-S0.3-COT-PAP | 1.75 | 1.75 | 1.75 | 5.00 | 7.04 | 5.73 |
| | 0shot-Text-S0.3-COT-TPD | 1.75 | 1.75 | 1.75 | 4.70 | 6.53 | 5.38 |
| | 0shot-Text-S0.3-CSV | 0.00 | 0.00 | 0.00 | 2.67 | 5.68 | 3.52 |
| | 0shot-Text-S0.3-PAP | 0.00 | 0.00 | 0.00 | 3.72 | 7.16 | 4.88 |
| | 0shot-Text-S0.3-TPD | 0.00 | 0.00 | 0.00 | 2.91 | 5.80 | 3.85 |
| | 0shot-Vision | 61.66 | 61.99 | 61.48 | 62.96 | 63.61 | 63.21 |
| | 0shot-Vision-COT | 59.35 | 59.19 | 59.16 | 59.58 | 59.66 | 59.61 |
| | 1shot-Text-S0.3 | 0.55 | 0.57 | 0.49 | 12.41 | 23.06 | 16.08 |
| | 1shot-Text-S0.3-COT | 0.50 | 0.50 | 0.45 | 11.47 | 18.85 | 14.16 |
| | 1shot-Vision | 55.43 | 54.91 | 54.51 | 58.14 | 58.29 | 58.21 |
| | 1shot-Vision-COT | 16.59 | 23.58 | 15.96 | 30.33 | 37.52 | 32.85 |
| GPT-4o-Mini | 0shot-Text | 0.00 | 0.00 | 0.00 | 3.88 | 6.95 | 4.97 |
| | 0shot-Text-S0.3 | 0.06 | 0.08 | 0.07 | 8.01 | 14.97 | 10.37 |
| | 0shot-Text-S0.3-COT | 2.75 | 2.75 | 2.75 | 8.98 | 14.20 | 10.75 |
| | 0shot-Text-S0.3-COT-CSV | 4.00 | 4.00 | 4.00 | 5.69 | 7.17 | 6.11 |
| | 0shot-Text-S0.3-COT-PAP | 9.50 | 9.50 | 9.50 | 10.38 | 10.68 | 10.48 |
| | 0shot-Text-S0.3-COT-TPD | 1.50 | 1.50 | 1.50 | 1.86 | 2.18 | 1.96 |
| | 0shot-Text-S0.3-CSV | 0.00 | 0.00 | 0.00 | 3.09 | 6.86 | 4.13 |
| | 0shot-Text-S0.3-PAP | 13.25 | 13.25 | 13.25 | 14.69 | 15.52 | 14.97 |
| | 0shot-Text-S0.3-TPD | 4.50 | 4.50 | 4.50 | 7.01 | 9.17 | 7.75 |
| | 0shot-Vision | 59.79 | 59.76 | 59.77 | 60.64 | 61.48 | 60.95 |
| | 0shot-Vision-COT | 59.69 | 59.68 | 59.67 | 60.12 | 60.59 | 60.29 |
| | 1shot-Text-S0.3 | 0.09 | 0.32 | 0.11 | 8.23 | 15.53 | 10.71 |
| | 1shot-Text-S0.3-COT | 0.17 | 0.75 | 0.26 | 10.02 | 17.56 | 12.71 |
| | 1shot-Vision | 22.10 | 23.80 | 22.23 | 33.83 | 44.08 | 37.45 |
| | 1shot-Vision-COT | 24.78 | 25.45 | 24.66 | 35.93 | 46.75 | 39.81 |
| Internvlm-76B | 0shot-Text | 5.87 | 15.45 | 6.70 | 12.10 | 19.43 | 14.55 |
| | 0shot-Text-S0.3 | 16.64 | 19.12 | 16.90 | 20.66 | 24.08 | 21.83 |
| | 0shot-Text-S0.3-COT | 11.75 | 11.75 | 11.75 | 13.39 | 14.79 | 13.86 |
| | 0shot-Text-S0.3-COT-CSV | 2.03 | 2.25 | 2.05 | 4.64 | 6.36 | 5.23 |
| | 0shot-Text-S0.3-COT-PAP | 15.85 | 16.26 | 15.86 | 16.59 | 17.41 | 16.87 |
| | 0shot-Text-S0.3-COT-TPD | 8.75 | 8.75 | 8.75 | 9.16 | 9.56 | 9.30 |
| | 0shot-Text-S0.3-CSV | 0.50 | 0.50 | 0.50 | 4.19 | 7.64 | 5.33 |
| | 0shot-Text-S0.3-PAP | 38.00 | 39.03 | 38.15 | 39.07 | 40.37 | 39.51 |
| | 0shot-Text-S0.3-TPD | 4.23 | 7.30 | 4.57 | 8.36 | 13.01 | 9.92 |
| | 0shot-Vision | 4.73 | 15.62 | 6.03 | 17.11 | 32.39 | 22.13 |
| | 0shot-Vision-COT | 11.99 | 15.10 | 12.59 | 25.63 | 39.81 | 30.49 |
| | 1shot-Text-S0.3 | 20.50 | 22.98 | 20.68 | 24.44 | 27.95 | 25.64 |
| | 1shot-Text-S0.3-COT | 38.12 | 37.71 | 36.67 | 43.13 | 43.61 | 43.32 |
| | 1shot-Vision | 33.08 | 37.78 | 31.98 | 40.59 | 43.70 | 41.67 |
| | 1shot-Vision-COT | 29.14 | 32.33 | 29.49 | 35.35 | 39.70 | 36.89 |
| Qwen | 0shot-Text | 0.00 | 0.00 | 0.00 | 0.00 | 0.00 | 0.00 |
| | 0shot-Text-S0.3 | 0.00 | 0.00 | 0.00 | 0.00 | 0.00 | 0.00 |
| | 0shot-Text-S0.3-COT | 3.00 | 3.00 | 3.00 | 3.12 | 3.24 | 3.16 |
| | 0shot-Text-S0.3-COT-CSV | 0.00 | 0.00 | 0.00 | 0.00 | 0.00 | 0.00 |
| | 0shot-Text-S0.3-COT-PAP | 6.75 | 6.75 | 6.75 | 6.79 | 6.83 | 6.80 |
| | 0shot-Text-S0.3-COT-TPD | 0.50 | 0.50 | 0.50 | 0.58 | 0.65 | 0.60 |

Table 6 – continued from previous page

| | | PRE | REC | F1 | affi PRE | affi REC | affi F1 |
|---|---|---|---|---|---|---|---|
| | 0shot-Text-S0.3-CSV | 0.00 | 0.00 | 0.00 | 0.00 | 0.00 | 0.00 |
| | 0shot-Text-S0.3-PAP | 12.00 | 12.00 | 12.00 | 12.00 | 12.00 | 12.00 |
| | 0shot-Text-S0.3-TPD | 1.25 | 1.25 | 1.25 | 1.29 | 1.33 | 1.30 |
| | 0shot-Vision | 4.17 | 27.94 | 6.29 | 16.61 | 30.00 | 21.21 |
| | 0shot-Vision-COT | 21.45 | 25.06 | 21.78 | 23.75 | 25.71 | 24.46 |
| | 1shot-Text-S0.3-COT | 0.00 | 0.00 | 0.00 | 0.00 | 0.00 | 0.00 |
| | 1shot-Vision | 7.96 | 25.63 | 9.18 | 23.34 | 33.78 | 27.03 |
| | 1shot-Vision-COT | 30.74 | 31.50 | 30.86 | 33.42 | 33.91 | 33.63 |
| Threshold | 0shot | 6.58 | 2.56 | 3.46 | 19.40 | 37.51 | 25.41 |

Table 7: Frequency anomalies in regular sine wave with extra noise

| | | PRE | REC | F1 | affi PRE | affi REC | affi F1 |
|---|---|---|---|---|---|---|---|
| Gemini-1.5-Flash | 0shot-Text | 3.49 | 4.33 | 3.42 | 36.32 | 32.22 | 32.67 |
| | 0shot-Text-S0.3 | 5.55 | 5.56 | 4.52 | 49.76 | 44.03 | 44.64 |
| | 0shot-Text-S0.3-COT | 1.02 | 1.54 | 0.99 | 10.87 | 10.11 | 10.06 |
| | 0shot-Text-S0.3-COT-CSV | 2.53 | 1.15 | 1.37 | 21.91 | 17.21 | 18.15 |
| | 0shot-Text-S0.3-COT-PAP | 4.24 | 2.95 | 3.18 | 23.96 | 16.49 | 18.35 |
| | 0shot-Text-S0.3-COT-TPD | 3.08 | 2.51 | 2.30 | 21.07 | 16.42 | 17.54 |
| | 0shot-Text-S0.3-CSV | 1.83 | 0.99 | 1.11 | 25.04 | 17.47 | 19.42 |
| | 0shot-Text-S0.3-PAP | 2.52 | 0.66 | 0.95 | 29.37 | 20.91 | 23.13 |
| | 0shot-Text-S0.3-TPD | 2.89 | 1.61 | 1.65 | 30.57 | 24.15 | 25.60 |
| | 0shot-Vision | 15.48 | 16.25 | 15.74 | 18.21 | 15.49 | 16.30 |
| | 0shot-Vision-COT | 15.02 | 15.58 | 15.12 | 17.98 | 15.41 | 16.14 |
| | 1shot-Text-S0.3 | 5.58 | 5.40 | 4.67 | 48.78 | 42.01 | 43.08 |
| | 1shot-Text-S0.3-COT | 6.38 | 6.62 | 5.62 | 48.53 | 39.45 | 41.23 |
| | 1shot-Vision | 9.43 | 9.56 | 9.22 | 24.40 | 23.19 | 22.68 |
| | 1shot-Vision-COT | 4.05 | 4.44 | 3.73 | 42.42 | 37.36 | 36.99 |
| GPT-4o-Mini | 0shot-Text | 3.07 | 3.90 | 2.91 | 34.43 | 28.86 | 29.92 |
| | 0shot-Text-S0.3 | 3.26 | 5.14 | 3.26 | 46.28 | 50.33 | 46.79 |
| | 0shot-Text-S0.3-COT | 2.94 | 4.54 | 2.84 | 25.88 | 24.39 | 24.13 |
| | 0shot-Text-S0.3-COT-CSV | 2.92 | 2.48 | 2.49 | 15.09 | 12.37 | 12.84 |
| | 0shot-Text-S0.3-COT-PAP | 5.28 | 5.28 | 5.28 | 7.74 | 7.07 | 7.27 |
| | 0shot-Text-S0.3-COT-TPD | 1.03 | 0.69 | 0.76 | 3.46 | 2.55 | 2.82 |
| | 0shot-Text-S0.3-CSV | 6.50 | 4.84 | 4.95 | 22.11 | 16.88 | 17.98 |
| | 0shot-Text-S0.3-PAP | 7.94 | 7.38 | 7.41 | 15.97 | 12.83 | 13.61 |
| | 0shot-Text-S0.3-TPD | 5.50 | 4.32 | 4.47 | 28.72 | 21.46 | 23.33 |
| | 0shot-Vision | 14.42 | 13.79 | 13.81 | 19.11 | 17.57 | 17.90 |
| | 0shot-Vision-COT | 14.69 | 13.79 | 14.00 | 18.20 | 16.15 | 16.66 |
| | 1shot-Text-S0.3 | 3.57 | 5.88 | 3.59 | 47.82 | 51.01 | 47.89 |
| | 1shot-Text-S0.3-COT | 4.36 | 7.62 | 4.50 | 46.82 | 53.21 | 48.43 |
| | 1shot-Vision | 11.36 | 9.50 | 8.97 | 46.21 | 40.67 | 40.96 |
| | 1shot-Vision-COT | 17.16 | 16.35 | 16.05 | 38.43 | 34.08 | 34.53 |
| Internvlm-76B | 0shot-Text | 8.05 | 15.96 | 8.36 | 19.88 | 22.41 | 19.82 |
| | 0shot-Text-S0.3 | 3.91 | 5.76 | 3.78 | 34.44 | 32.29 | 31.51 |
| | 0shot-Text-S0.3-COT | 4.95 | 5.62 | 5.02 | 13.32 | 13.13 | 12.71 |
| | 0shot-Text-S0.3-COT-CSV | 2.15 | 2.22 | 1.76 | 17.98 | 15.28 | 15.60 |
| | 0shot-Text-S0.3-COT-PAP | 4.71 | 5.32 | 4.70 | 8.91 | 8.33 | 8.32 |
| | 0shot-Text-S0.3-COT-TPD | 4.54 | 4.64 | 4.38 | 11.81 | 9.53 | 10.09 |
| | 0shot-Text-S0.3-CSV | 3.32 | 4.13 | 2.69 | 31.42 | 25.17 | 26.43 |

Table 7 – continued from previous page

| | | PRE | REC | F1 | affi PRE | affi REC | affi F1 |
|---|---|---|---|---|---|---|---|
| | 0shot-Text-S0.3-PAP | 6.93 | 18.54 | 7.33 | 20.58 | 25.28 | 21.37 |
| | 0shot-Text-S0.3-TPD | 3.54 | 7.06 | 3.58 | 20.54 | 17.54 | 17.58 |
| | 0shot-Vision | 6.42 | 27.41 | 8.37 | 35.89 | 49.66 | 39.36 |
| | 0shot-Vision-COT | 6.67 | 13.42 | 7.40 | 27.07 | 30.84 | 27.36 |
| | 1shot-Text-S0.3 | 4.04 | 4.93 | 3.62 | 32.27 | 31.25 | 30.26 |
| | 1shot-Text-S0.3-COT | 2.52 | 2.45 | 1.95 | 32.61 | 28.65 | 28.26 |
| | 1shot-Vision | 4.07 | 17.07 | 4.29 | 42.30 | 44.61 | 40.25 |
| | 1shot-Vision-COT | 5.65 | 9.66 | 5.86 | 39.17 | 40.44 | 37.75 |
| Qwen | 0shot-Text | 0.00 | 0.00 | 0.00 | 0.00 | 0.00 | 0.00 |
| | 0shot-Text-S0.3 | 0.00 | 0.00 | 0.00 | 0.00 | 0.00 | 0.00 |
| | 0shot-Text-S0.3-COT | 1.29 | 1.37 | 1.31 | 2.12 | 2.06 | 2.06 |
| | 0shot-Text-S0.3-COT-CSV | 0.00 | 0.00 | 0.00 | 0.00 | 0.00 | 0.00 |
| | 0shot-Text-S0.3-COT-PAP | 1.05 | 1.05 | 1.05 | 1.90 | 1.48 | 1.60 |
| | 0shot-Text-S0.3-COT-TPD | 0.30 | 0.41 | 0.30 | 1.33 | 1.12 | 1.18 |
| | 0shot-Text-S0.3-CSV | 0.00 | 0.00 | 0.00 | 0.00 | 0.00 | 0.00 |
| | 0shot-Text-S0.3-PAP | 3.25 | 3.25 | 3.25 | 3.88 | 3.65 | 3.73 |
| | 0shot-Text-S0.3-TPD | 0.86 | 1.01 | 0.85 | 1.75 | 1.56 | 1.59 |
| | 0shot-Vision | 3.49 | 52.89 | 5.61 | 31.02 | 56.16 | 39.71 |
| | 0shot-Vision-COT | 5.07 | 10.77 | 5.34 | 8.26 | 11.14 | 9.27 |
| | 1shot-Text-S0.3-COT | 0.00 | 0.00 | 0.00 | 0.00 | 0.00 | 0.00 |
| | 1shot-Vision | 4.43 | 16.33 | 4.87 | 41.37 | 47.88 | 41.80 |
| | 1shot-Vision-COT | 2.41 | 4.67 | 2.66 | 11.85 | 15.40 | 12.97 |
| Threshold | 0shot | 3.34 | 3.17 | 2.90 | 43.61 | 80.70 | 56.47 |

Table 8: Point noises anomalies in regular sine wave with Gaussian noise

| | | PRE | REC | F1 | affi PRE | affi REC | affi F1 |
|---|---|---|---|---|---|---|---|
| Gemini-1.5-Flash | 0shot-Text | 1.74 | 2.13 | 1.59 | 25.29 | 29.59 | 26.65 |
| | 0shot-Text-S0.3 | 3.16 | 3.24 | 2.52 | 42.84 | 43.91 | 42.33 |
| | 0shot-Text-S0.3-COT | 2.14 | 2.36 | 2.04 | 12.24 | 13.54 | 12.42 |
| | 0shot-Text-S0.3-COT-CSV | 1.36 | 1.44 | 1.12 | 15.91 | 17.39 | 16.20 |
| | 0shot-Text-S0.3-COT-PAP | 3.20 | 3.01 | 3.05 | 18.53 | 17.87 | 17.73 |
| | 0shot-Text-S0.3-COT-TPD | 1.41 | 1.29 | 1.24 | 14.45 | 15.35 | 14.58 |
| | 0shot-Text-S0.3-CSV | 0.56 | 0.20 | 0.21 | 18.01 | 18.43 | 17.80 |
| | 0shot-Text-S0.3-PAP | 0.52 | 0.36 | 0.40 | 20.46 | 21.62 | 20.60 |
| | 0shot-Text-S0.3-TPD | 1.76 | 1.44 | 1.48 | 22.54 | 25.53 | 23.44 |
| | 0shot-Vision | 16.40 | 23.95 | 17.81 | 41.99 | 41.51 | 40.64 |
| | 0shot-Vision-COT | 52.07 | 74.07 | 57.06 | 87.58 | 80.69 | 82.49 |
| | 1shot-Text-S0.3 | 1.98 | 2.32 | 1.74 | 42.93 | 43.62 | 42.06 |
| | 1shot-Text-S0.3-COT | 4.45 | 6.45 | 4.16 | 41.18 | 41.31 | 40.00 |
| | 1shot-Vision | 51.76 | 71.87 | 56.92 | 89.71 | 86.76 | 87.30 |
| | 1shot-Vision-COT | 35.73 | 55.60 | 40.53 | 74.40 | 72.92 | 72.70 |
| GPT-4o-Mini | 0shot-Text | 1.44 | 2.33 | 1.40 | 27.13 | 30.86 | 28.14 |
| | 0shot-Text-S0.3 | 4.18 | 8.57 | 4.54 | 42.21 | 52.73 | 46.02 |
| | 0shot-Text-S0.3-COT | 3.29 | 3.06 | 2.72 | 22.44 | 25.08 | 23.00 |
| | 0shot-Text-S0.3-COT-CSV | 3.21 | 2.97 | 2.98 | 12.96 | 14.78 | 13.43 |
| | 0shot-Text-S0.3-COT-PAP | 9.50 | 9.50 | 9.50 | 11.31 | 11.20 | 11.20 |
| | 0shot-Text-S0.3-COT-TPD | 0.31 | 0.26 | 0.26 | 3.83 | 4.08 | 3.86 |
| | 0shot-Text-S0.3-CSV | 7.82 | 7.78 | 7.78 | 20.13 | 21.64 | 20.42 |
| | 0shot-Text-S0.3-PAP | 13.85 | 13.77 | 13.78 | 20.05 | 20.37 | 20.09 |

Table 8 – continued from previous page

| | | PRE | REC | F1 | affi PRE | affi REC | affi F1 |
|---|---|---|---|---|---|---|---|
| | 0shot-Text-S0.3-TPD | 4.58 | 4.57 | 4.57 | 21.37 | 22.99 | 21.72 |
| | 0shot-Vision | 41.91 | 45.92 | 41.57 | 76.05 | 74.43 | 74.48 |
| | 0shot-Vision-COT | 41.55 | 45.81 | 41.43 | 72.90 | 71.14 | 71.31 |
| | 1shot-Text-S0.3 | 4.14 | 8.09 | 4.28 | 42.12 | 52.09 | 45.60 |
| | 1shot-Text-S0.3-COT | 5.64 | 10.55 | 5.99 | 40.98 | 52.15 | 44.92 |
| | 1shot-Vision | 33.29 | 37.83 | 33.19 | 71.85 | 72.18 | 70.90 |
| | 1shot-Vision-COT | 33.30 | 39.88 | 33.51 | 72.22 | 72.06 | 70.86 |
| Internvlm-76B | 0shot-Text | 13.25 | 21.58 | 13.63 | 23.04 | 28.73 | 24.93 |
| | 0shot-Text-S0.3 | 6.17 | 7.62 | 6.08 | 23.42 | 26.53 | 24.26 |
| | 0shot-Text-S0.3-COT | 9.65 | 10.44 | 9.68 | 16.69 | 18.60 | 17.37 |
| | 0shot-Text-S0.3-COT-CSV | 3.67 | 4.17 | 3.47 | 17.26 | 18.61 | 17.49 |
| | 0shot-Text-S0.3-COT-PAP | 7.76 | 9.23 | 7.83 | 12.08 | 13.10 | 12.40 |
| | 0shot-Text-S0.3-COT-TPD | 6.76 | 7.53 | 6.85 | 11.60 | 12.34 | 11.84 |
| | 0shot-Text-S0.3-CSV | 1.71 | 1.40 | 1.29 | 24.40 | 26.54 | 24.70 |
| | 0shot-Text-S0.3-PAP | 11.81 | 23.53 | 12.45 | 25.24 | 34.09 | 28.20 |
| | 0shot-Text-S0.3-TPD | 5.19 | 8.47 | 5.46 | 18.19 | 20.99 | 19.13 |
| | 0shot-Vision | 16.95 | 47.68 | 22.03 | 55.46 | 65.17 | 58.43 |
| | 0shot-Vision-COT | 6.29 | 14.72 | 7.42 | 34.55 | 41.95 | 36.87 |
| | 1shot-Text-S0.3 | 7.38 | 8.30 | 7.16 | 24.13 | 27.35 | 25.17 |
| | 1shot-Text-S0.3-COT | 12.16 | 11.92 | 10.70 | 34.66 | 34.30 | 33.14 |
| | 1shot-Vision | 10.80 | 20.32 | 11.11 | 45.91 | 49.07 | 45.36 |
| | 1shot-Vision-COT | 8.32 | 13.05 | 8.37 | 37.04 | 42.25 | 38.01 |
| Qwen | 0shot-Text | 0.00 | 0.00 | 0.00 | 0.00 | 0.00 | 0.00 |
| | 0shot-Text-S0.3 | 0.00 | 0.00 | 0.00 | 0.00 | 0.00 | 0.00 |
| | 0shot-Text-S0.3-COT | 1.30 | 1.31 | 1.30 | 1.83 | 1.80 | 1.80 |
| | 0shot-Text-S0.3-COT-CSV | 0.00 | 0.00 | 0.00 | 0.00 | 0.00 | 0.00 |
| | 0shot-Text-S0.3-COT-PAP | 2.50 | 2.50 | 2.50 | 3.03 | 3.02 | 3.02 |
| | 0shot-Text-S0.3-COT-TPD | 1.25 | 1.25 | 1.25 | 1.46 | 1.47 | 1.46 |
| | 0shot-Text-S0.3-CSV | 0.00 | 0.00 | 0.00 | 0.00 | 0.00 | 0.00 |
| | 0shot-Text-S0.3-PAP | 7.50 | 7.50 | 7.50 | 8.11 | 8.09 | 8.10 |
| | 0shot-Text-S0.3-TPD | 1.00 | 1.00 | 1.00 | 2.48 | 2.53 | 2.47 |
| | 0shot-Vision | 4.76 | 45.49 | 6.90 | 28.33 | 49.37 | 35.65 |
| | 0shot-Vision-COT | 12.68 | 18.82 | 13.03 | 16.26 | 19.68 | 17.43 |
| | 1shot-Text-S0.3-COT | 0.00 | 0.00 | 0.00 | 0.00 | 0.00 | 0.00 |
| | 1shot-Vision | 7.18 | 25.56 | 8.21 | 41.94 | 52.88 | 45.10 |
| | 1shot-Vision-COT | 11.38 | 14.62 | 11.00 | 25.47 | 28.22 | 25.98 |
| Threshold | 0shot | 2.98 | 2.87 | 2.38 | 37.16 | 70.83 | 48.67 |

Table 9: Trend anomalies, but no negating trend, and less noticeable speed changes

| | | PRE | REC | F1 | affi PRE | affi REC | affi F1 |
|---|---|---|---|---|---|---|---|
| Gemini-1.5-Flash | 0shot-Text | 0.00 | 0.00 | 0.00 | 4.82 | 9.69 | 6.43 |
| | 0shot-Text-S0.3 | 8.07 | 8.51 | 7.84 | 16.98 | 20.53 | 18.27 |
| | 0shot-Text-S0.3-COT | 2.69 | 3.02 | 2.74 | 5.24 | 7.76 | 6.10 |
| | 0shot-Text-S0.3-COT-CSV | 0.25 | 0.25 | 0.25 | 2.03 | 3.59 | 2.54 |
| | 0shot-Text-S0.3-COT-PAP | 3.75 | 3.75 | 3.75 | 5.23 | 6.53 | 5.65 |
| | 0shot-Text-S0.3-COT-TPD | 1.50 | 1.50 | 1.50 | 3.26 | 4.69 | 3.76 |
| | 0shot-Text-S0.3-CSV | 0.00 | 0.00 | 0.00 | 2.76 | 6.33 | 3.78 |
| | 0shot-Text-S0.3-PAP | 0.00 | 0.00 | 0.00 | 3.33 | 7.14 | 4.52 |
| | 0shot-Text-S0.3-TPD | 0.00 | 0.00 | 0.00 | 3.79 | 7.75 | 5.08 |

Table 9 – continued from previous page

| | | PRE | REC | F1 | affi PRE | affi REC | affi F1 |
|---|---|---|---|---|---|---|---|
| | 0shot-Vision | 57.50 | 57.50 | 57.50 | 57.50 | 57.50 | 57.50 |
| | 0shot-Vision-COT | 57.50 | 57.50 | 57.50 | 57.50 | 57.50 | 57.50 |
| | 1shot-Text-S0.3 | 11.30 | 12.34 | 10.77 | 21.27 | 24.65 | 22.52 |
| | 1shot-Text-S0.3-COT | 12.40 | 13.31 | 12.12 | 23.69 | 27.45 | 25.23 |
| | 1shot-Vision | 57.91 | 57.91 | 57.89 | 57.99 | 58.00 | 57.99 |
| | 1shot-Vision-COT | 16.24 | 22.09 | 15.77 | 25.75 | 30.56 | 27.48 |
| GPT-4o-Mini | 0shot-Text | 1.25 | 1.25 | 1.25 | 5.49 | 9.74 | 6.88 |
| | 0shot-Text-S0.3 | 0.11 | 0.43 | 0.17 | 11.15 | 21.05 | 14.56 |
| | 0shot-Text-S0.3-COT | 0.81 | 0.97 | 0.85 | 2.19 | 3.50 | 2.63 |
| | 0shot-Text-S0.3-COT-CSV | 5.25 | 5.25 | 5.25 | 7.55 | 10.35 | 8.32 |
| | 0shot-Text-S0.3-COT-PAP | 6.75 | 6.75 | 6.75 | 7.42 | 7.75 | 7.55 |
| | 0shot-Text-S0.3-COT-TPD | 0.25 | 0.25 | 0.25 | 0.52 | 0.77 | 0.61 |
| | 0shot-Text-S0.3-CSV | 0.25 | 0.25 | 0.25 | 3.46 | 8.72 | 4.83 |
| | 0shot-Text-S0.3-PAP | 11.25 | 11.25 | 11.25 | 12.41 | 12.91 | 12.59 |
| | 0shot-Text-S0.3-TPD | 6.50 | 6.50 | 6.50 | 8.67 | 10.87 | 9.39 |
| | 0shot-Vision | 57.50 | 57.50 | 57.50 | 57.50 | 57.50 | 57.50 |
| | 0shot-Vision-COT | 57.25 | 57.25 | 57.25 | 57.25 | 57.25 | 57.25 |
| | 1shot-Text-S0.3 | 0.52 | 1.10 | 0.65 | 11.20 | 21.18 | 14.61 |
| | 1shot-Text-S0.3-COT | 1.84 | 2.06 | 1.91 | 12.11 | 20.06 | 14.94 |
| | 1shot-Vision | 57.50 | 57.50 | 57.50 | 57.50 | 57.50 | 57.50 |
| | 1shot-Vision-COT | 57.00 | 57.00 | 57.00 | 57.00 | 57.00 | 57.00 |
| Internvlm-76B | 0shot-Text | 10.36 | 16.13 | 10.94 | 15.48 | 21.61 | 17.51 |
| | 0shot-Text-S0.3 | 26.61 | 27.85 | 25.85 | 32.78 | 34.41 | 33.37 |
| | 0shot-Text-S0.3-COT | 10.67 | 12.35 | 10.81 | 12.60 | 14.33 | 13.19 |
| | 0shot-Text-S0.3-COT-CSV | 5.55 | 6.25 | 5.59 | 8.85 | 10.68 | 9.49 |
| | 0shot-Text-S0.3-COT-PAP | 15.11 | 15.75 | 15.19 | 16.03 | 16.70 | 16.26 |
| | 0shot-Text-S0.3-COT-TPD | 8.89 | 9.63 | 8.98 | 9.67 | 10.54 | 9.96 |
| | 0shot-Text-S0.3-CSV | 5.09 | 5.39 | 5.14 | 9.83 | 13.88 | 11.21 |
| | 0shot-Text-S0.3-PAP | 33.80 | 35.78 | 34.00 | 35.62 | 37.63 | 36.24 |
| | 0shot-Text-S0.3-TPD | 3.99 | 6.55 | 4.19 | 6.64 | 9.57 | 7.62 |
| | 0shot-Vision | 34.58 | 42.52 | 35.03 | 40.75 | 47.54 | 43.03 |
| | 0shot-Vision-COT | 51.88 | 51.66 | 51.72 | 52.85 | 53.70 | 53.14 |
| | 1shot-Text-S0.3 | 29.16 | 30.57 | 28.69 | 36.38 | 37.31 | 36.77 |
| | 1shot-Text-S0.3-COT | 39.77 | 41.00 | 39.16 | 46.03 | 46.61 | 46.29 |
| | 1shot-Vision | 37.95 | 42.16 | 37.07 | 44.67 | 46.94 | 45.50 |
| | 1shot-Vision-COT | 51.37 | 53.02 | 51.53 | 54.02 | 55.26 | 54.47 |
| Qwen | 0shot-Text | 0.00 | 0.00 | 0.00 | 0.00 | 0.00 | 0.00 |
| | 0shot-Text-S0.3 | 0.00 | 0.00 | 0.00 | 0.00 | 0.00 | 0.00 |
| | 0shot-Text-S0.3-COT | 0.00 | 0.00 | 0.00 | 0.00 | 0.00 | 0.00 |
| | 0shot-Text-S0.3-COT-CSV | 0.00 | 0.00 | 0.00 | 0.00 | 0.00 | 0.00 |
| | 0shot-Text-S0.3-COT-PAP | 0.00 | 0.00 | 0.00 | 0.00 | 0.00 | 0.00 |
| | 0shot-Text-S0.3-COT-TPD | 0.00 | 0.00 | 0.00 | 0.00 | 0.00 | 0.00 |
| | 0shot-Text-S0.3-CSV | 0.00 | 0.00 | 0.00 | 0.00 | 0.00 | 0.00 |
| | 0shot-Text-S0.3-PAP | 0.00 | 0.00 | 0.00 | 0.00 | 0.00 | 0.00 |
| | 0shot-Text-S0.3-TPD | 0.00 | 0.00 | 0.00 | 0.00 | 0.00 | 0.00 |
| | 0shot-Vision | 4.71 | 21.94 | 6.04 | 13.05 | 22.47 | 16.24 |
| | 0shot-Vision-COT | 21.65 | 22.75 | 21.76 | 22.20 | 22.76 | 22.40 |
| | 1shot-Text-S0.3 | 0.00 | 0.00 | 0.00 | 0.00 | 0.00 | 0.00 |
| | 1shot-Text-S0.3-COT | 0.00 | 0.00 | 0.00 | 0.00 | 0.00 | 0.00 |
| | 1shot-Vision | 12.72 | 28.32 | 13.09 | 28.20 | 38.21 | 31.73 |
| | 1shot-Vision-COT | 29.98 | 31.18 | 29.79 | 34.14 | 35.79 | 34.78 |

Table 10: Yahoo S5 dataset

|  |  | PRE | REC | F1 | affi PRE | affi REC | affi F1 |
|---|---|---|---|---|---|---|---|
| Gemini-1.5-Flash | 0shot-Text | 0.18 | 3.16 | 0.26 | 16.10 | 18.50 | 16.46 |
|  | 0shot-Text-S0.3 | 20.21 | 22.73 | 20.10 | 31.04 | 30.41 | 29.91 |
|  | 0shot-Text-S0.3-CSV | 3.82 | 3.95 | 3.82 | 13.80 | 10.71 | 11.60 |
|  | 0shot-Text-S0.3-PAP | 16.48 | 16.67 | 16.51 | 26.52 | 23.85 | 24.53 |
|  | 0shot-Text-S0.3-TPD | 14.97 | 16.62 | 15.11 | 28.32 | 27.09 | 27.00 |
|  | 0shot-Vision | 30.44 | 45.17 | 31.74 | 54.82 | 57.25 | 55.77 |
|  | 1shot-Text-S0.3 | 20.41 | 21.93 | 20.40 | 31.39 | 30.66 | 30.32 |
|  | 1shot-Vision | 28.54 | 31.06 | 28.56 | 45.27 | 43.10 | 43.64 |
| GPT-4o-Mini | 0shot-Text | 2.74 | 4.04 | 2.50 | 15.26 | 14.68 | 14.40 |
|  | 0shot-Text-S0.3 | 36.02 | 37.42 | 36.06 | 41.38 | 41.57 | 41.26 |
|  | 0shot-Text-S0.3-CSV | 18.28 | 18.48 | 18.29 | 27.40 | 25.31 | 25.86 |
|  | 0shot-Text-S0.3-PAP | 25.62 | 25.70 | 25.63 | 29.31 | 28.48 | 28.70 |
|  | 0shot-Text-S0.3-TPD | 4.17 | 4.65 | 4.22 | 11.98 | 11.00 | 11.20 |
|  | 0shot-Vision | 50.88 | 57.84 | 51.31 | 71.42 | 74.07 | 72.19 |
|  | 1shot-Text-S0.3 | 39.74 | 41.03 | 39.73 | 46.59 | 45.90 | 45.76 |
|  | 1shot-Vision | 17.22 | 18.07 | 17.18 | 32.81 | 34.18 | 32.61 |
| Internvlm-76B | 0shot-Text | 30.42 | 37.91 | 30.54 | 38.31 | 42.25 | 39.53 |
|  | 0shot-Text-S0.3 | 46.08 | 47.28 | 46.10 | 48.13 | 48.47 | 48.13 |
|  | 0shot-Text-S0.3-CSV | 6.30 | 8.49 | 6.34 | 16.59 | 14.92 | 15.32 |
|  | 0shot-Text-S0.3-PAP | 53.72 | 54.52 | 53.76 | 55.43 | 55.64 | 55.49 |
|  | 0shot-Text-S0.3-TPD | 14.80 | 17.51 | 14.86 | 19.96 | 21.27 | 20.29 |
|  | 0shot-Vision | 12.42 | 29.87 | 13.52 | 35.19 | 40.91 | 37.41 |
|  | 1shot-Text-S0.3 | 50.17 | 52.32 | 50.21 | 52.12 | 53.23 | 52.46 |
|  | 1shot-Vision | 1.84 | 11.19 | 2.47 | 20.62 | 23.15 | 20.96 |
| Isolation-Forest | 0shot | 2.07 | 29.26 | 3.25 | 16.75 | 33.18 | 21.94 |
| Qwen | 0shot-Text | 0.00 | 0.00 | 0.00 | 0.00 | 0.00 | 0.00 |
|  | 0shot-Text-S0.3 | 0.00 | 0.00 | 0.00 | 0.00 | 0.00 | 0.00 |
|  | 0shot-Text-S0.3-CSV | 0.00 | 0.00 | 0.00 | 0.00 | 0.00 | 0.00 |
|  | 0shot-Text-S0.3-PAP | 19.89 | 19.89 | 19.89 | 19.91 | 19.90 | 19.91 |
|  | 0shot-Text-S0.3-TPD | 2.18 | 2.18 | 2.18 | 2.19 | 2.21 | 2.20 |
|  | 0shot-Vision | 3.10 | 19.21 | 3.40 | 16.19 | 25.37 | 19.53 |
|  | 1shot-Text-S0.3 | 0.00 | 0.00 | 0.00 | 0.00 | 0.00 | 0.00 |
|  | 1shot-Vision | 3.39 | 5.68 | 3.24 | 18.37 | 21.61 | 19.22 |
| Threshold | 0shot | 3.99 | 21.99 | 5.51 | 16.40 | 31.80 | 21.03 |

# E MATHEMATICAL FORMULATIONS OF HYPOTHESES 2 AND 3

## E.1 HYPOTHESIS 2: REPETITION BIAS AND PERIODICITY DETECTION

Let $f(t)$ be a time series and $T(f)$ be its tokenized representation in an LLM's vocabulary space $\mathcal{V}$. We define:

1) Perfect periodicity: $f(t + P) = f(t)$ for some period $P > 0$

2) Noisy periodicity: $f(t+P) = f(t) + \epsilon(t)$ where $\epsilon(t) \sim \mathcal{N}(0, \sigma^2)$ and $\sigma \ll \min_{t,t'} |f(t) - f(t')|$

Note that while noisy periodicity is defined on numerical values, the tokenization process $T(\cdot)$ maps these values to discrete tokens, making perfect periodicity in token space impossible for noisy periodic signals.

Let $A(f)$ be the LLM's anomaly detection accuracy on time series $f$ and $\mathcal{D}_P$ be the set of all periodic time series with period $P$. Given there exists an optimal anomaly detector $B$, whose accuracy is $B^*(f)$, that can achieve near-perfect accuracy on both perfect and noisy periodic signals. The hypothesis states:

For any $f_1, f_2 \in \mathcal{D}_P$, if $T(f_1)$ exhibits token-level periodicity and $T(f_2)$ does not, then:

$$E[A(f_1)] \gg E[A(f_2)]$$

while the optimal detector maintains consistent performance:

$$B^*(f_1) \approx B^*(f_2) \approx 1$$

This formulation suggests that LLMs' performance difference is due to token-level repetition bias rather than the inherent complexity of the anomaly detection task, as a properly designed detector can achieve near-perfect performance on both cases.

## E.2 HYPOTHESIS 3: ARITHMETIC ABILITY AND PATTERN RECOGNITION

Let $M$ be an LLM and $M'$ be the same LLM with impaired arithmetic ability.

1) Define arithmetic ability $\alpha(M)$ as accuracy on basic arithmetic tasks:

$$\alpha(M) = E_{x,y}[\Vdash[M(\text{"What is } x + y\text{?"}) = x + y]]$$

2) Define reasoning ability $\rho(M)$ as accuracy on non-arithmetic reasoning tasks:

$$\rho(M) = E_{q \in \mathcal{Q}}[\Vdash[M(q) = \text{correct}]]$$

where $\mathcal{Q}$ is a set of logical reasoning questions.

3) Obtain $M'$ by training $M$ on incorrect arithmetic examples while preserving reasoning:

$$\begin{cases} \alpha(M') \ll \alpha(M) \\ \rho(M') \approx \rho(M) \end{cases}$$

4) Hypothesis holds if:

$$E_{f \in \mathcal{D}}[A_M(f)] \gg E_{f \in \mathcal{D}}[A_{M'}(f)]$$

where $\delta$ is small and dataset $\mathcal{D}$ is arbitrary. To falsify the hypothesis, we show the difference is negligble or reversed for certain datasets.

