# OpenReview forum: "Can LLMs Understand Time Series Anomalies?"
_ICLR.cc/2025/Conference — ICLR 2025 Poster_

### Official Review · Reviewer_hKqp · 2024-10-28

**Soundness:** 2
**Presentation:** 3
**Contribution:** 2
**Rating:** 5
**Confidence:** 4

**Summary:**

This study investigates large language models' (LLMs) understanding of time-series anomalies by proposing and testing specific hypotheses.

**Strengths:**

The study incorporates a wide range of prompting and representation techniques.

**Weaknesses:**

Writing
1. Some redundancy is present; for instance, the discussion in Section 3.3 lacks direct relevance to problem motivation and is only tangentially related.

Experiment/Study
1. Some hypotheses are tested without adequate scientific rigor. For example, in the discussion of Hypothesis (7) on architecture bias, several variables—such as model architecture (e.g., QWEN vs. GPT-4o-mini), model size, and the content and size of the pretraining corpus—are not controlled. This lack of control makes it difficult to attribute performance differences specifically to architecture, as they may instead stem from variations in model size or pretraining data.

2. No mention of using statistical tests for hypothesis. For instance, in the "retrained Hypothesis 1 on CoT reasoning," it is unclear if the claimed result is from statistical testing with a null hypothesis of no difference or heuristic. It would be made the study scientific and convincing by reporting some results from, say t-test on accuracy difference through bootstrapping.

**Questions:**

See weaknesses

---

> ### Author Response · Authors · 2024-11-27
>
> Thank you for your in-depth review. Here is our point-by-point response:
>
> **1. Section 3.3's Relevance:**
>
> *the discussion in Section 3.3 lacks direct relevance to problem motivation and is only tangentially related...*
>
> Our work builds directly on Gruver et al.'s (2023) findings about LLMs' "broad pattern extrapolation capabilities" in time series analysis. While their work focused on forecasting, we demonstrate that these capabilities are crucial for anomaly detection, as both tasks rely on understanding and extrapolating temporal patterns. Section 3.3 establishes this theoretical bridge, providing the foundation for testing these assumptions in zero-shot anomaly detection settings.
>
> **2. Model Architecture Analysis:**
>
> *model size, and the content and size of the pretraining corpus—are not controlled ...*
>
> We acknowledge the limitations in controlling for pretraining corpus and model size when comparing architectures. Like Nielsen et al. (2024), who analyzed encoder vs decoder models without pretraining from scratch, we face similar computational resource constraints that prevent training models from scratch. We have revised our claim from "LLMs' time series understanding vary significantly across different model architectures" to "across different model families" to better reflect our findings' scope. Hypothesis 7's results demonstrate specialization patterns across modalities and anomaly types, rather than pure architectural effects.
>
> **3. Statistical Testing:**
>
> *No mention of using statistical tests for hypothesis ...*
>
> It is very challenging to conduct comprehensive statistical testing in LLM evaluation for the following reasons:
>
> - API rate limits and costs make extensive bootstrapping impractical
> - Proprietary models restrict access to full token distribution probabilities (at most logprob of top 5 tokens)
> - Deterministic inference in many models limits traditional statistical approaches
>
> However, we want to note that we have done our best to pursue reliability through:
>
> - Running each experiment three times
> - Reporting mean performance with reasonable standard deviations (typically within 1% of mean)
> - Documenting all experimental conditions for reproducibility
>
> [1] Gruver et al. (2023). Large Language Models Are Zero-Shot Time Series Forecasters. arXiv:2310.07820
>
> [2] Nielsen et al. (2024) "Encoder vs Decoder: Comparative Analysis of Encoder and Decoder Language Models on Multilingual NLU Tasks." arXiv:2406.13469

---

> > ### Comment · Reviewer_hKqp · 2024-12-01
> > **Discussion Response**
> >
> > Thank you for your thoughtful response.
> >
> > $\textbf{W1}$: Thank you for specifying your motivation to include this particular discussion. While I think the paragraph serves as a motivation for your work, I am not fully convinced this section offers 'theoretical bridge' as no proof or empirical study is done here.
> >
> >
> >
> > $\textbf{W2}$:  Thank you for reconsidering your hypothesis 7 and providing further explanation. However, I respectfully maintain that there are still numerous variables that need to be controlled to validly test this claim. Each model differs significantly in terms of its text and image components (when applicable), as well as its size and pretraining corpus. While the cited article provides a clear and focused study (e.g., encoder vs. decoder models), its methodology does not appear to translate directly to your setting due to the lack of control over influencing factors.
> >
> >
> > All in all, may I ask, then, what the intended takeaway is in stating that models from different sources demonstrate varying capabilities? What is the significance of this observation? Wouldn’t such differences already be expected given the models’ varied performance on widely available benchmarks?
> >
> >
> > My primary concern on this point remains that some claims in the paper may require additional evidence or consideration. I believe this perspective is shared by reviewers iE2G and M4GK.
> >
> > 	•	For example, when stating that certain results vary "significantly," it would be valuable to provide a formal statistical test to support this conclusion.
> > 	•	Similarly, when claiming that something is "too subtle to be visually detected by humans but becomes apparent when computing the moving average of the gradient," in hypothesis 5, is there human evaluation to validate this observation? Does this claim hold across varying resolutions in the vision component (e.g., using different image qualities for models like OpenAI’s VLMs)?
> >
> >
> >
> > $\textbf{W3}$: I understand the challenges inherent to this type of analysis. My primary concern lies in the limited number of anomalies generated at a similar scale, which may make the observed randomness in anomalies less significant. For instance, as outlined in Section B.1, the point anomalies appear to be generated based on a sine wave bounded between -1 and 1. This suggests that candidate models are primarily tested on a small subset of anomalies.
> >
> >
> >
> > Overall I appreciate your effort to address my concern. I personally like this array of study. However I do think the rigor of the study can be improved, as discussed above. I remain my score.

---

### Official Review · Reviewer_M4GK · 2024-11-01

**Soundness:** 2
**Presentation:** 2
**Contribution:** 2
**Rating:** 6
**Confidence:** 5

**Summary:**

The paper explores the potential of using Large Language Models (LLMs) for analyzing anomalies in time-series data, focusing on the insights in terms of reasoning that can be gained from detecting them. To do this, the paper proposes a set of hypotheses and scenarios to assess the effectiveness of LLMs when processing time-series anomalies, providing an overview of their strengths and limitations, as well as the possible parallels with human reasoning.

**Strengths:**

- It introduces an attractive problem related to assessing how LLMs can detect time-series anomalies and explores whether there are similarities in how humans approach this task.
- It emphasizes the importance of analyzing time-series data from different perspectives, such as using visual features in a multi-modal setting, which may yield better results than relying solely on text input.
- The paper is generally well-written and relatively easy to follow.

**Weaknesses:**

- The contributions of the paper are not explicitly defined, making the goal of the paper ambiguous. For example, it is unclear whether the goal was to provide a strategy for assessing how LLMs deal with anomaly detection (via the hypotheses), or to simply present the results of evaluating LLMs under specific settings (as highlighted in the abstract and introduction).
- The paper lacks a clearly defined and replicable method. The reasoning for establishing the hypotheses needs stronger support from previous work, and the evaluation design should clearly outline how each hypothesis is assessed and the logic behind it.
- The experiments lack essential details, such as the differences between prompt variants, and the results are inconsistent across all evaluated settings (e.g., combining zero-shot and few-shot in the same figures). This raises doubts about whether the highlighted findings are supported by the experimental results.
- The paper contains claims like "prevailing beliefs about LLMs" or "expected patterns" that are not properly discussed. For instance, the Related Work section should explain what these prevailing beliefs are, and the Definitions section should clearly define what is considered an expected pattern.
- The definitions of time-series are not formalized and are ambiguous. Many of them are not used, leaving it unclear as to why they were included. Additionally, there are inconsistencies between them (see details below).

# Detailed Comments
## Introduction
- There is not clear justification of focusing the evaluation solely in the F1 score
- The content is mostly focused on the experimental findings, so it is not clear what is the paper contribution
## Related Work
- It is mostly a literature review, with no discussion over the main topic of reasoning over times-series anomalies
- There is no discussion about the continuously mentioned controversy about time-series analytics in LLMs. A paper close to that topic, Jin et al. (2024a), was not discussed in this section.
- The claim of lack for visual M-LLM on time series is not accurate. For instance, Jin et al. (2024a) already included references to that approach

## Definitions
- Definition of anomaly is very ambiguous. What is an expected pattern?
- There is no definition of the multiple thresholds
- Equation 1 needs a better explanation. What is G? A forecaster? If G generates x_t, so where the external x_t comes from?
- The anomaly detection algorithm is not clear. For example, it is a function that generates anomaly scores for each x_t...
- Interval-based anomalies definition seems incorrect. Why i /in {1,...,k} for a given y_t? It is probably Y, although it seems an unused definition
- If the interest in the data-point level anomalies, why using interval-based
- The few-shot definition is inconsistent with previous definitions and g is undefined. The output is probably referring to Y, instead of the actual s_1 or y_1
- The definition of zero-shot needs more details, contextualized to the prompt setting

## Time-Series Categories
- Shuffled time series? It is contractionary to the concept of series and time-dependency
- Trend example is not that clear. Which gradient?
- Frequency example is not clear, there is no significant shift in Fig. 1(b)
- What is an expected pattern? It is a very ambiguous term used frequently

## Time-Series Forecasting
- There are not clear details about the implication of extrapolation: "Therefore, both time series forecasting, and anomaly detection rely heavily on extrapolation."
- Connection with of time series and tokens needs to be clearly specified

## Understanding LLM
- "To demystify LLMs’ anomaly detection capabilities" What are the myths?
- There are many assumptions underlying the description of a LLM from a cognitive science perspective. The argument needs to be carefully elaborated because it is implying that a LLM behaves like a human brain, which is a strong statement. Then, it is not clear how the evaluation between reflexible and reflective modes is conducted
- It is not clear how the second hypothesis may be evaluated using noisy time series
- There is necessary to include some support for proposing the fourth and fifth hypotheses
- Fifth hypothesis evaluation is not as consistent with the approach of the referred paper
- In general, the hypothesis is not aligned with the paper goal regarding anomalies and are mostly focused on looking for some parallels with human reasoning

## Prompting
- The prompts seem very important, but there is no information about how they are designed. For instance, it is not clear what the variants are doing

## Experiments
- The process for generating of data sets does not provide design details, even in the Appendix. It should disclose more technical details, and show some examples, since it is over this data that the hypotheses will be evaluated
- Metrics depend on a threshold, but there is no mention of it
- It is not clear what Figures 2,4,5,6,7,8 are showing. It is a comparison between prompts that may or may not be comparable and there is not clear point of reference
- There is not consistency between the selected prompts, so the results are hardly comparable between anomaly types or LLM engines. Also combining zero-shot and few-shot prompts will introduce unnecessary noise to the evaluation
- In general, the experiments are not providing enough support for evaluating the hypotheses. For example, with the third hypothesis, how the arithmetic capability is removed? The experiment needs to include all this type of details
- The visual-text experiment, which is a major claim for the paper, need to explore with more details, how it was conducted, how the prompts were designed to achieve a fair comparison
- Why the frequency anomalies failed? It needs a better explanation, perhaps showing an example of the actual task that the LLM is doing and the chart with the time-series

**Questions:**

- What are the prevailing beliefs about LLMs when processing time series? Where had they come from?
- How to impair a LLM for evaluating the third hypothesis?
- What are the foundations and support to evaluate a LLM from the perspective of cognitive science?


Post rebuttal comments:
Thank you for the authors' rebuttals, which address many of the questions. I thus increase the score.

---

> ### Author Response · Authors · 2024-11-27
>
> Thank you for your time and efforts to write a very detailed and thorough review. While we respectfully differ on some technical interpretations, we appreciate the opportunity to address each concern in detail.
>
> There are major misunderstandings of our work. We would like to address five key aspects of your review:
>
> **Clarity of Purpose and Contribution**: The paper's focuses on assessment strategy and results do not contradict each other. Specifically, our paper 1. Formulates testable hypotheses based on current literature claims 2. Designs controlled experiments to isolate specific aspects of LLM behavior 3. Report results that retain (not necessarily support) or falsify hypotheses. It's meaningless to just showcase possible experiment design without results - the scientific value emerges from their integration.
>
> **Methodological Soundness**: Comprehensive prompt examples and technical details are provided in the Appendix (pages 16-17), with sufficient information for replication. The code repository is also available, making the methodology fully transparent and replicable.
>
> **Experimental Design Validity**: The combination of different prompts in one figure actually strengthens the evaluation by showing best achievable performance across all variants, avoiding the common benchmarking pitfall of selective reporting. This approach provides a more complete and fair comparison while accounting for model-specific limitations.
>
> **Definitions of "Prevailing Beliefs":** The "prevailing beliefs" about LLMs' capabilities stem from recent literature [Gruver et al., 2023; Jin et al., 2024] claiming their "broad pattern extrapolation capabilities" and "interpretable predictions." We have modified the paper to explicitly define these terms.
>
> **Time Series Definition:** We have clarified these definitions in the updated version.
>
> Let us elaborate on each of these points in more detail.

---

> > ### Author Response · Authors · 2024-11-27
> >
> > ### Details
> >
> > *... with no discussion over the main topic of reasoning over times-series anomalies*
> >
> > This reflects a gap in existing literature rather than a limitation of our review. To our knowledge, no previous work has specifically investigated reasoning mechanisms in time-series understanding with LLM. While some papers like Tan (2024) discuss reasoning, they focus primarily on performance improvements rather than examining how LLMs reason about time series. Our work aims to fill this crucial gap by analyzing the reasoning process itself.
> >
> > *"The paper lacks a clearly defined and replicable method... differences between prompt variants..."*
> >
> > We respectfully disagree. The prompt design examples are provided in Appendix A.4 with specific examples. Our empirical evidence for "how to impair an LLM" is detailed in A.4 and A.5, showing that "the integer addition accuracy drops from 100% to 12%, and the floating point addition accuracy drops from 100% to 45.0%." We provide comprehensive examples enabling replication, with our codebase available.
> >
> > *"The process for generating of data sets does not provide design details, even in the Appendix. It should disclose more technical details, and show some examples..."*
> >
> > We have provided comprehensive technical details through Algorithm 1 and Appendix B.1, which include both the detailed algorithm and specific hyperparameters. We have included illustrated examples in Figure 1, and for clarity, we have offered more more illustrative examples in Appendix B in our revision.
> >
> > *"The claim of lack of visual M-LLM on time series is not accurate. For instance, Jin et al. (2024a) already include a reference to that approach..."*
> >
> > We need to clarify two points regarding this comment:
> >
> > 1. If you're referring to Jin et al. 's Time-LLM approach: While they do employ patch embedding for time series, this differs fundamentally from visual M-LLMs in architecture and training paradigm. Visual M-LLMs incorporate complete vision transformer architectures and benefit from extensive pretraining on diverse image-text pairs, enabling knowledge transfer from visual understanding tasks. Jin et al.'s approach has nothing to do to visualize the time series.
> > 2. If you're referring to the visual methods referenced in Jin et al.: Upon careful examination, none of the cited works actually address time series:
> >    - Misra et al. (2023) focuses on cross-domain image tasks like classification and texture recognition
> >    - Tsimpoukelli et al. (2021) addresses image captioning
> >    - Ma et al. (2023) discusses speech emotion recognition
> >
> > All evidence supports our claim on the lack of work in VLLM for time series.
> >
> > *"There is not consistency between the selected prompts... combining zero-shot and few-shot prompts will introduce unnecessary noise..."*
> >
> > We respectfully disagree with this assessment. Separating zero-shot and few-shot results could actually obscure true comparative performance. Consider a common scenario in model evaluation: Model A achieves 0.85 with 1-shot but 0.3 with 0-shot prompting, while Model B achieves 0.8 and 0.4 respectively. Separate performance reports with different prompting styles (e.g., 0-shot) can provide an incomplete or misleading picture of overall model capabilities. Not combining performance in one plot makes it difficult for the audience to see what's the best scenario under certain settings.
> >
> > Our approach aligns with established practices in the field. For instance, the widely-referenced MMLU benchmark (https://paperswithcode.com/sota/multi-task-language-understanding-on-mmlu) reports each model's best achievable performance. Our methodology is even more comprehensive, as we evaluate all models across all possible prompting styles, avoiding cases where model performance was not optimized by the original authors (e.g., DBRX 132B) and **reducing noise.**
> >
> > This comprehensive evaluation becomes particularly important when considering model-specific limitations. For example, some models like InternVLM could have incomplete multi-round chat support (documented in OpenGVLab/InternVL#51). By showing all variants together, we ensure fair comparison while accounting for such technical constraints, ultimately providing a more complete and accurate assessment of model capabilities.

---

> > > ### Author Response · Authors · 2024-11-27
> > >
> > > *"There are not clear details about the implication of extrapolation..."*
> > >
> > > Our work builds on assumptions from Gruver et al. (2023), who claimed LLMs possess "broad pattern extrapolation capabilities" in time series analysis. While their work focused on forecasting, we argue these capabilities are crucial for anomaly detection, as both tasks rely on understanding and extrapolating temporal patterns. This connection provides the theoretical foundation for testing their assumptions in zero-shot anomaly detection.
> > >
> > > *"Definition of anomaly is very ambiguous. What is an expected pattern?"*
> > >
> > > Expected pattern is the governing function we expect the time series to follow. For example, when observing 1, 1, 2, 3, 5, 8, 11, the expected pattern is the recurrence relation y_t = y_{t-1} + y_{t-2}. Data-point level anomalies occur when specific segments deviate from this pattern, in contrast to sequence-level anomalies where the entire sequence deviates.
> > >
> > > *"Metrics depend on a threshold, but there is no mention of it"*
> > >
> > > The evaluation uses affinity F1 scores, which are parameter-free by design [Huet et al., 2022]. We do not see why we need thresholds.
> > >
> > > *"Shuffled time series? It is contractionary to the concept of series and time-dependency"*
> > >
> > > Time series anomaly detection encompasses two types of anomalies: those detectable through point-wise statistical properties (identifiable even when shuffled) and those emerging from temporal dependencies (requiring preserved ordering). This dual nature is well-established, as evidenced by Lai et al. (2023) and Tan et al. (2024). Shuffling the time series is a good sanity check to see if a model is picking up the positional representation of the time series.
> > >
> > > *"Trend example is not that clear. Which gradient?"*
> > >
> > > We appreciate this request for clarity regarding gradient calculations. The gradient refers to the first-order difference between adjacent points:
> > >
> > > Gradient = (Value₂ - Value₁) / (Time₂ - Time₁)
> > >
> > > However, recognizing that this basic calculation can be sensitive to local fluctuations, usually we want to implement a more robust trend analysis using smoothed gradients. This involves:
> > >
> > > Computing a moving average to reduce noise:
> > >
> > > 1. MA(t) = (Value(t) + Value(t-1) + ... + Value(t-w+1)) / w
> > >
> > > Calculating the gradient using these smoothed values:
> > >
> > > 1. Gradient(t) = (MA(t+1) - MA(t)) / (Time(t+1) - Time(t))
> > >
> > > As illustrated in Figure 8, provides a robust trend estimation and can be used to perform anomaly detection based on gradient deviation. We will incorporate these technical specifications into the manuscript.
> > >
> > > *... from a cognitive science perspective. The argument needs to be carefully elaborated because it is implying that a LLM behaves like a human brain, which is a strong statement...*
> > >
> > > This is an unfair judgment. We never claim "a LLM behaves like a human brain''. Perspectives from cognitive science provide inspiration to systematically examine LLM behavior patterns in time-series analysis tasks. This approach has many parallels. For example, recent work in Nature Human Behaviour [Strachan et al., 2024] explicitly examines "the possibility that these models exhibit behaviour that is indistinguishable from human behaviour in theory of mind tasks." As demonstrated by Vafa et al. (2024), investigating these parallels provides insights into both LLM capabilities and human cognition.
> > >
> > > *"Why the frequency anomalies failed?"*
> > >
> > > This consistent failure across different model families suggests a fundamental limitation rather than implementation-specific issues. When presented with this example frequency anomalies (e.g.,[ https://i.imgur.com/M57hrXP.png](https://i.imgur.com/M57hrXP.png)), all tested LLMs fail to identify the anomaly despite clear deviations. A detailed investigation of this systematic failure warrants its own dedicated study.
> > >
> > > *"There is necessary to include some support for proposing the fourth and fifth hypotheses"*
> > >
> > > Our fourth and fifth hypotheses about visual representation in time series analysis are supported by emerging research trends. For instance, Dong et al. (2024) explore similar themes, proposing prompts that explicitly reference visual thinking (e.g., "think about the visual representation") even in pure language model contexts.
> > >
> > > *"The visual-text experiment, which is a major claim for the paper, need to explore with more details"*
> > >
> > > We included 13 text prompts and 9 vision prompts to ensure fair visual-text comparison. As detailed in Appendix A.1, we run sanity checks to ensure vision components don't impair text performance. For reproducibility, we emphasize using the vision-model even for pure text experiments. If there are specific details that we missed in our investigation, we would appreciate your suggestions.

---

> > > > ### Author Response · Authors · 2024-11-27
> > > >
> > > > References:
> > > >
> > > > - Gruver et al. (2023). Large Language Models Are Zero-Shot Time Series Forecasters. arXiv:2310.07820
> > > > - Tan et al. (2024). Are Language Models Actually Useful for Time Series Forecasting? arXiv:2406.16964
> > > > - Lai et al. (2023). Nominality Score Conditioned Time Series Anomaly Detection by Point/Sequential Reconstruction. NeurIPS 2023
> > > > - Strachan et al. (2024). Testing theory of mind in large language models and humans. Nature Human Behaviour
> > > > - Vafa et al. (2024). Do Large Language Models Perform the Way People Expect? arXiv:2406.01382
> > > > - Jin et al. (2024). Position: What Can Large Language Models Tell Us about Time Series Analysis. arXiv:2402.02713
> > > > - Huet et al. (2022). Local evaluation of time series anomaly detection algorithms. KDD Conference
> > > > - Dong et al. (2024). Can LLMs Serve As Time Series Anomaly Detectors?. *arXiv preprint arXiv:2408.03475*.

---

### Official Review · Reviewer_CZBV · 2024-11-04

**Soundness:** 3
**Presentation:** 3
**Contribution:** 3
**Rating:** 6
**Confidence:** 4

**Summary:**

This paper investigates whether MM-LLMs can effectively detect anomalies in time series data,from the perspective of validating 7 hypotheses about LLM performance on the proposed tasks. By conducting controlled experiments, the authors assess LLM capabilities across various anomaly types and input representations (text vs. visual). The findings reveal that LLMs perform better with visualised time series data. Additionally, the study challenges several assumptions about LLMs, including their reliance on arithmetic abilities and repetition biases, which were previously thought to aid in pattern recognition, and the impact of CoT on time series tasks. Finally the authors show that different model architectures can have significantly different performance in anomaly detection, suggesting that previous results using just one or two architectures may not generalise.

**Strengths:**

S1. The authors articulate hypotheses proposed by existing literature or propose some based on analyses of current research and then conduct controlled, synthetic, experiments to test these hypotheses.

S2. Visual vs. Textual Representation: this paper extends existing research into the feasibility of visual detection models for time series analysis by comparing multimodal LLMs' performance on visual versus textual time series data, revealing that visual inputs often yield better anomaly detection. Given the applicability of recent multimodal LLMs for a lot of tasks, the analysis on time series anomaly detection is timely.

S3. The study sets up interesting tests for the hypothesis around arithmetic abilities and repetitive biases. The noise injection and artificial arithmetic reasoning drop seem like a well-constructed approach.

**Weaknesses:**

W1. While the study is insightful, its primary focus on hypothesis testing rather than direct practical application might limit its immediate utility for practitioners looking to deploy LLMs in anomaly detection systems. This can still be helpful for theoretical progress and providing direction for further time-series anomaly detection research, however I think some of the experiments were not defined very strictly or in such a way that they could be built upon for future analysis which hampers the benefit of the work.

W2. The experiments and the hypothesis validation, primarily rely on synthetic datasets for testing. Introducing real-world datasets, such as the 18 datasets found in the TSB-UAD univariate anomaly detection benchmark (Paparrizos et. al. 2022) can help with further validating these hypotheses in real-world settings.

**Questions:**

Q1. For further validation of the arithmetic ability hypothesis, did the authors consider simply comparing different variations of the same model architectures (e.g. by size) and their performance on standard arithmetic benchmarks and correlating it with performance on anomaly detection? It seems that for many of these hypothesis relying on other real-world benchmark proxies could provide additional verification.

Q2. For validating hypothesis 5, the authors claim the test dataset “is too subtle to be visually detected by humans”. How is this validated?

Q3. I did appreciate that the authors stuck by the claim that humans preferred visual anomaly detection and also presented most of their results visually, however this did make it somewhat difficult to interpret the results. Would it be possible to summarise the most important tests for each hypothesis in a single table? I think this could add to the presentation.

Nits:

P.6. “We hypotheses…” grammar.

---

> ### Author Response · Authors · 2024-11-27
>
> Thank you for your thorough and encouraging review. We particularly appreciate your recognition of our constructed experimental design and insights into multimodal LLMs for time series tasks. Let us address each point:
>
> **Immediate Utility (W1):**
>
> *I think some of the experiments were not defined very strictly ...*
>
> Thank you for the suggestions, we have implemented rigorous formulations for key hypotheses - particularly hypothesis 2 (repetition) and 3 (arithmetic). Detailed results are provided in Appendix E.
>
> **Real-world Validation (W2)**
>
> *... such as the 18 datasets found in the TSB-UAD univariate anomaly detection benchmark (Paparrizos et. al. 2022) can help ...*
>
> We appreciate the suggestion to include TSB-UAD benchmark datasets. However, as noted by Wu et al. (2021), some benchmark datasets (e.g., NASA, Yahoo) contain anomalies that even human experts cannot identify without external context. Their lack of anomaly description also makes CoT variants inapplicable. Our synthetic datasets were specifically designed to isolate and test distinct anomaly types. Nevertheless, we agree that real-world validation would strengthen our findings. We have conducted additional experiments using the Yahoo dataset, with results included in Appendix D Table 10.
>
> **Questions:**
>
> *... different variations of the same model architectures (e.g. by size) ?*
>
> Q1: While model size comparison is valuable, the landscape of suitable open-source multimodal models was limited during our study. Only Qwen-VL 7B and Intern-VL2 series could perform anomaly detection tasks adequately. Smaller models like Phi-3-vision-128k (3.8B parameters) proved insufficient, and commercial LLMs lack access to size variants with controlled training conditions. With recent releases of Qwen2-VL and multimodal LLaMA 3.2, we are conducting additional experiments with comparable model sizes.
>
> *How is this (visual subtlety) validated?*
>
> Q2: We have carried out a small visual subtlety validation experiment involving five participants evaluating twenty randomly sampled "subtle" time series alongside controlled samples. Results showed detection rates of 0.0 / 20.0 and 17.2 / 20.0 respectively. The experiment is also backed up by Alexandra et al. (2016). We acknowledge the need for a larger participant pool and will expand this study.
>
> [1] Wu, R., & Keogh, E. J. (2021). Current time series anomaly detection benchmarks are flawed and are creating the illusion of progress. IEEE transactions on knowledge and data engineering, 35(3), 2421-2429.
>
> [2] Alexandra S. Mueller and Brian Timney. (2016) Visual acceleration perception for simple and complex motion patterns. PLoS ONE, 11(2):e0149413. ISSN 1932-6203. doi: 10.1371/journal. pone.0149413.

---

> > ### Comment · Reviewer_CZBV · 2024-12-02
> >
> > Thank you for your responses to all the reviewers. I appreciate the effort put in to validate the hypotheses put forward in the paper and to follow-up on all the questions. The discussion around the usefulness of existing 'real-world' benchmarks was helpful context and the addition of the yahoo dataset evaluations adds to the paper.
> >
> > With regards to the Volume Under Surface (VUS) metric mentioned by reviewer 'nLhj': the authors can consider taking the probability of the token as the score and using that for the metric. LLM evaluations do not need to only generate binary labels. However I understand the limitations of using an API to generate some of these evaluations may make this infeasible.
> >
> > Reading through the other reviews and responses I most agree with reviewer 'hKqp' in stating that 'I personally like this array of study. However I do think the rigor of the study can be improved'.
> >
> > I will maintain my score: I think as is this work does provide some contribution and can be accepted, however further refinement could make it an even better paper.

---

### Official Review · Reviewer_nLhj · 2024-11-05

**Soundness:** 3
**Presentation:** 3
**Contribution:** 3
**Rating:** 3
**Confidence:** 4

**Summary:**

With the increasing popularity of LLMs and applications to various tasks, such as forecasting, their performance for anomaly detection remains largely unexplored. This work studies zero-shot and few-shot scenarios of popular LLMs and formulates hypotheses to debunk current norms in this fastly emerging field. The analysis suggests that several beliefs were wrong so this paper offers a useful tool for helping to move the field forward.

**Strengths:**

S1. Timely problem to understand the performance of LLMs in this context
S2. Reasonable hypotheses formulated and tested
S3. Interesting findings useful for the community

**Weaknesses:**

W1. Unclear focus on LLMs and not pure focus on time-series Foundation Models.
W2. Missing evaluation measures
W3. Missing benchmarks
W4. Missing experimental comparison against baselines to understand the performance of these methods

**Questions:**

W1. Unclear focus on LLMs and not pure focus on time-series Foundation Models.

Indeed, the field started by changing language models, assuming their sequential structure could have some benefit for time series as well. I think from the first moment, the time-series community understood how problematic that was.. and the lack of strong experimental results, comparisons with baselines, etc. exacerbated the situation and more and more folks tried to use such LLMs.. however, very fast, the field moved into pure time-series foundation models and abandoned LLM or VLMs for such use. So even though I appreciate the effort, at the same time I think the picture is already much much bigger to just make claims about LLMs which are obviously problematic for this area. I think a much bigger study to include VLMs/LLMs/FM for this area is needed

W2. Missing evaluation measures

The study focuses on problematic solutions for assessing anomaly detectors. There has been significant recent progress that is omitted [a].

[a] "Volume under the surface: a new accuracy evaluation measure for time-series anomaly detection." Proceedings of the VLDB Endowment 15.11 (2022): 2774-2787.

W3. Missing benchmarks

Similarly, limited datasets are used when established benchmark exists for this problem [b]

[b] "TSB-UAD: an end-to-end benchmark suite for univariate time-series anomaly detection." Proceedings of the VLDB Endowment 15.8 (2022): 1697-1711.

W4. Missing experimental comparison against baselines to understand the performance of these methods

Even though I understand this might be somewhat out of scope, I think what matters in the end is how they performs vs. SOTA methods. Let's say even if they greatly understand time-series, in the end where they stand in comparison to SOTA techniques. Therefore, an evaluation using such methods ([a,b] should be a good starting point) is needed as well

---

> ### Author Response · Authors · 2024-11-27
>
> Thank you for your review highlighting four important areas for improvement in our work. We appreciate the opportunity to clarify our positions and address each concern in detail.
>
> *... the field moved into pure time-series foundation models and abandoned LLM*
>
> **Focus on LLMs vs Foundation Models**: Our work fills a crucial gap by providing a methodological critique of LLMs for time series, carefully examining existing assumptions rather than purely empirical results (which have been accomplished by Tan et. al 2024). The timing is particularly relevant as the community remains divided on whether specialized time-series foundation models will ultimately prove as useful as LLM with other modalities.
>
> This skepticism is supported by recent observations - time series foundational models have not yet demonstrated significant improvements over existing approaches (e.g., Chronos's forecasting performance remains relatively flat), and in some cases, like TimeGPT, can be outperformed by well-tuned traditional models like ARIMA in long-term forecasting tasks. [2] Our analysis helps inform this ongoing debate about the fundamental capabilities and limitations of different approaches to time series analysis.
>
> *... There has been significant recent progress that is omitted [a].*
>
> **Evaluation Metrics:** Regarding the Volume Under Surface (VUS) metric [a], we note that this metric is not applicable to our LLM evaluation framework as LLMs generate binary labels rather than continuous anomaly scores for each timestamp. Our choice of metrics aligns with the discrete nature of LLM outputs.
>
> *... limited datasets are used when established benchmark*
>
> **Benchmark Datasets:** While TSB-UAD [b] is a comprehensive benchmark, it includes known problematic datasets (e.g., NASA, Yahoo, as noted in Wu et al. 2021) where some anomalies are even not recognizable by human experts without external information. For example, in the Yahoo dataset (Figure 17, bottom left), we observe arbitrary endpoint selections in anomaly regions. Additionally, TSB-UAD datasets lack descriptions of abnormal patterns, making it impossible to test Chain-of-Thought prompting without extensive manual labeling. We deliberately chose synthetic datasets to ensure clear anomaly definitions and eliminate mislabeling issues. These well-controlled experiments guarantee that our hypothesis testing produces valid results. However, we acknowledge the value of real-world validation and are actively conducting additional experiments with carefully selected real-world datasets. We have performed extra experiments with the relatively short Yahoo dataset and include it in the Appendix D Table 10, where results show the tested LLM approach outperforms traditional methods.
>
> *... in the end where they stand in comparison to SOTA techniques.*
>
> **Baseline Comparisons:** While comprehensive SOTA comparisons were not our primary focus, we do provide relevant performance comparisons in Appendix, Observation 8. In the experiment, we demonstrate that it outperforms traditional algorithms. We acknowledge that additional comparisons could provide valuable context and will consider expanding this in future work.
>
> [1] Ansari et al. (2024). Chronos: Learning the language of time series. arXiv:2403.07815.
>
> [2] https://medium.com/@claywr/time-gpt-for-forecasting-kicking-the-tires-6dd564dae62f
>
> [3] Wu, R., & Keogh, E. J. (2021). Current time series anomaly detection benchmarks are flawed and are creating the illusion of progress. *IEEE transactions on knowledge and data engineering*, *35*(3), 2421-2429.

---

> > ### Comment · Reviewer_nLhj · 2024-12-03
> >
> > I would like to thank the authors for the feedback. Unfortunately, I am not convinced about several statements.
> >
> > For example, it's unclear why VUS cannot work on your setting and raises the question if the particular output in any way biases results vs. what the community has been using until now.
> >
> > TSB-UAD is the largest heterogeneous dataset in this area. There are about 2000 time series, there are synthetic and artificial generation mechanisms to help assess more complex cases or control the anomaly injection process. Wu et al. 2021 raise concerns, but it does not necessarily justify the exclusion of datasets. Why is triviality a problem for example? Shouldn't a complex method work on easy/trivial anomalies? Why such datasets should be removed? Why they haven't been removed from the UCR classification archive when the simplest possible method achieves 100% accuracy? We are in a continuous space, so claiming mislabeling issues is a stretch when we don't even know where an anomaly starts and ends. Some mislabling issues (lag/alignments) are solved with appropriate evaluation measures. Other mislabelling issues are unclear if they cause any problems. You mentioned anomalies that not even human experts can detect. Great, we agree, so no method identifies them and that does not impact scores for methods (all get credit or not get credit for example). In every argument, there can be a counterargument and there is no clear consensus in the community. For example, Wu et al. 2021 proposed a dataset with only a single anomaly per time series, when every other dataset in this area contains more than one anomaly per time series. There are methods favoring the single anomaly setting for example.

---

### Official Review · Reviewer_iE2G · 2024-11-05

**Soundness:** 3
**Presentation:** 3
**Contribution:** 3
**Rating:** 6
**Confidence:** 4

**Summary:**

This paper studies how Large Language Models (LLMs) reason about anomalous time series. The authors formulate 7 hypothesis, based on recent work at the intersection of time series & LLMs, their own understanding of the problem, and prevailing hypotheses about LLMs, and carefully design experiments using synthetic time series to accept or refute their hypotheses.

**Strengths:**

1. The intersection of time series and LLMs is an exciting and active area of research. The authors present a very neat study on LLM capabilities on the anomaly detection problem.
2. The hypotheses are interesting and most experiments are designed well.
3. The paper is well-written, and aesthetically appealing, and some findings are interesting.
4. The authors demonstrate a fairly good understanding of the literature, on time series, anomaly detection and LLMs.

**Weaknesses:**

1. **Some claims need to be supported or softened:** The authors make some claims in the paper which are not well supported by the experiments and the overall story. (a) *on the question of LLMs understanding time series*:  There are many different notions of understanding in addition to anomaly detection, such as in predictive tasks such as classification [1] and forecasting [2], but also other inductive tasks such as understanding of (granger) causality [3]. The ability to reason about one class of problems, does not imply the ability to reason about other classes of problems, hence LLMs are expected to do well in some tasks, and not in some others. What the authors show evaluate is if LLMs can reason about some kinds of anomalies using synthetic data. In the abstract, the authors mention: "These insights pave the way for more effective LLM-based approaches in time series analysis". However, it is unclear how some insights, while interesting, are actionable (e.g., on perceptual reasoning abilities). I must note that some insights are actionable, e.g. on long context bias. (c) The model-specific finding is not new, and is consistent with prior work [4, 5] which has established the need and value of model selection for anomaly detection.
2. **On the defining anomaly detection:** Anomaly detection is a tricky problem with a lot of nuances. The authors only study one variant of the problem, i.e. detecting *4 different kinds of anomalies in synthetic time series with labels in zero-shot and few-shot settings*. Meanwhile, there is a large body of work in unsupervised or semi-supervised anomaly detection, which does not rely on explicitly labeled anomalies as training examples.  Finally, the authors state (line 223 onwards) the connections between forecasting and anomaly detection. These are valid. However, (1) only a subset of AD methods rely on forecasting, other methods rely on reconstruction or differences in latent space [4], (2) the authors do not rely on their LLMs ability to "forecast", and therefore the ability that they are testing have little to do with extrapolating, as against interpolating based on past and future context.
3. **Reproducibility**: The authors haven't shared examples from their generated dataset, or the inference and evaluation code. They also have not shared visual examples of their generated time series, which makes it harder to reproduce their work.
4. **Hypothesis 5 ("LLMs’ understanding of anomalies is consistent with human visual perception.") is too broad and not well supported:** (a) The study by Mueller & Timney, 2016, only exhibits 1 form of human perceptual bias. Disproving it only shows that LLMs do not have this particular bias. (b) It is not immediately clear to me how this bias manifests in anomaly detection. I am not sure how constant speed change differs from acceleration (which causes a change in speed). This hypothesis would greatly benefit from being fully specified, motivated, and examples of datasets shown.
5. **Hypothesis 2 (LLMs’ repetition bias (Holtzman et al., 2020) corresponds precisely to its ability to identify
and extrapolate periodic structure in the time series. is unclear:** The hypothesis has 3 key terms, "identify", "extrapolate" and "periodic", but the authors do not vary any of these key terms. The settings that the authors consider have nothing to do with "extrapolation" (forecasting), they only consider periodic or periodic time series with noise (which is still periodic), and the paper provides no evidence if or how well LLMs identify periodic structure, and how that corresponds to their findings. I think examples of the generated dataset and the nature of the noise would benefit this section. Also, correlating the ability of models to identify / extrapolate periodic structure with their ability to identity anomalies in periodic and non-periodic time series, would be a much better way to break down this complex claim in my opinion.
6. **Hypothesis 1 ("LLMs do not benefit from engaging in step-by-step reasoning about time series data.") is too broad:** The authors only provide evidence that the models are not capable of chain of thought reasoning. CoT reasoning is only one way to do step-by-step reasoning. The hypothesis should instead be that these models are not helped by engaging is CoT reasoning or some variant of this claim.
7. **Use of Images:** This is an interesting finding and is consistent with recent work [3]. I wonder if highlighting the location of the anomaly in the image, just like the authors have done it for illustrative purposes would benefit AD performance even more. I suspect it might be true because I am not convince that models understand this statement: `[{"start": 171, "end": 178}]` especially because the reader has no idea how the time series plots look, if they have an x and y-axis that's marked with sufficient font size and precision.
8. **Synthetic data:** I understand why the authors use synthetic data, but without knowledge of the difficulty of the synthetic dataset, and how SoTA and simpler methods perform on these datasets. Also from the tables in the Appendix it seems that a simple threshold-based baseline does really well. This raises the question about the limited nature of anomalies, for example [5] can inject many more varieties of anomalies to real-world signals, which provides a much larger, and diverse set of real-world anomaly examples than the considered work. Also the authors cite [6] but do not use their datasets for their experiments. This might be important given that this studies' primary focus is on measuring LLM's ability to detect anomalies.

### Minor
1. Line 297 -> Hypotheses -> Hypothesize
2. Line 335, Missing .
3. Line 532, To -> to


### References
1. Cai, Yifu, et al. "Jolt: Jointly learned representations of language and time-series." Deep Generative Models for Health Workshop NeurIPS 2023. 2023.
2. Gruver, Nate, et al. "Large language models are zero-shot time series forecasters." Advances in Neural Information Processing Systems 36 (2024).
3. Cai, Yifu, et al. "TimeSeriesExam: A time series understanding exam." arXiv preprint arXiv:2410.14752 (2024).
4. Schmidl, Sebastian, Phillip Wenig, and Thorsten Papenbrock. "Anomaly detection in time series: a comprehensive evaluation." Proceedings of the VLDB Endowment 15.9 (2022): 1779-1797.
5. Goswami, Mononito, et al. "Unsupervised model selection for time-series anomaly detection." arXiv preprint arXiv:2210.01078 (2022).
6. Wu, Renjie, and Eamonn J. Keogh. "Current time series anomaly detection benchmarks are flawed and are creating the illusion of progress." IEEE transactions on knowledge and data engineering 35.3 (2021): 2421-2429.

**Questions:**

Please see above.

---

> ### Author Response · Authors · 2024-11-27
>
> Thank you for your thorough and encouraging review. We particularly appreciate your recognition of our work's contributions to time series analysis and LLMs, and the experimental design quality. Let us address each weakness:
>
> *Claims need to be supported or softened*
>
> We agree our claims about "LLM understanding" should be more specific: our findings do relate specifically to anomaly detection capabilities. We acknowledge other aspects like classification [1] and causality [3] should be backed with other set of experiments. However, to falsify some previous assumptions (like the hypothesis 1) made based on the LLM's extrapolation capability, we still only need the one counterexample. We remove broad claims about “paving the way”.
>
> *Defining anomaly detection is tricky*
>
> We'll revise our framing to acknowledge the broader landscape. Our work specifically studies unsupervised detection where anomaly labels are unavailable. While previous work [2] connects forecasting and anomaly detection, our experiments focus on detecting pattern deviations using full context, rather than predicting future values.
>
> *Reproducibility concerns*
>
> We have prepared a public repository with example synthetic datasets with visualizations, complete evaluation code and detailed experimental settings. We have added visualizations of the dataset used in Appendix B. For the evaluation code, we use the same code as in affinity F1 paper, and we note that it is parameter free. For the experimental settings, we provide detailed examples in Appendix A4 and A5.
>
> *Hypothesis 1 & 5 is too broad*
>
> For Hypothesis 5, our study specifically examines one perceptual bias documented by Mueller & Timney, and we have provided visual examples of the datasets in Appendix B of the updated paper. Regarding the broadness of the hypotheses, we understand this limitation and had taken a modest approach in our claims. In Hypothesis 1, we had carefully phrased our findings as "no evidence is found" rather than making absolute claims of impossibility, and specifically mentioned it is only about“explicit reasoning prompts through Chain of Thought” in the original pdf. While we acknowledge these hypotheses could be more narrowly defined, we aimed to balance between specificity and broader implications.
>
> *Hypothesis 2 needs clarification*
>
> The definitions are as follows:
>
> - "Identify": ability to recognize recurring patterns with period T in time series (e.g., daily cycles in power consumption)
> - "Extrapolate": predict the next k values following the history events
> - "Periodic": strict mathematical definition where f(t+T) = f(t) for some T>0
>
> For better understanding, we provide a more rigorous definition in Appendix E. The examples of periodic vs noisy periodic series are presented in Appendix B.

---

> > ### Comment · Reviewer_iE2G · 2024-11-30
> > **Thanks for your responses!**
> >
> > Dear Authors,
> >
> > Thank you so much for your responses, I appreciate it!
> > I would recommend making changes to the paper in a different ink so they are easier to track during rebuttal.
> > Also, unfortunately, I could not find examples for Hypothesis 5.
> >
> > Overall, I like the work, I think it is interesting.
> >
> > **Recommendations**:
> > 1. Please improve the reproducibility of the work by sharing their code and datasets publicly.
> > 2. I would also recommend that the authors mention the caveats discussed during the rebuttal in the paper, and at length in the appendix.
> > 3. For a paper, which treats LLMs as a scientific experiment, the rigor of experiments should also be scientific, and as such the hypotheses should be well defined. I understand balancing specificity and broader implications, perhaps the authors could consider having specific results in the Results section and having the broader implications in the Conclusion?
> >
> > **Impressions & Questions**:
> > 1. Unfortunately, Hypothesis 5 still does not make sense to me.
> > 2. I would like to know the authors' thoughts on (7) Use of Images and (8) Synthetic Data
> >
> > I would like to stay with my score. Good luck!

---

### Author Response · Authors · 2024-11-27

We are deeply grateful to all reviewers for their constructive feedback. In our updated manuscript, we have made several improvements, including:

- Added example visualizations of anomaly patterns in Appendix B
- Expanded experimental validation using the Yahoo dataset (Appendix D)
- Strengthened definitions regarding interval-based anomalies and anomaly types
- Clarified our hypotheses with more precise definitions in Appendix E

While we appreciate Reviewer M4GK's detailed comments, we respectfully note some misunderstandings of our methodology - particularly regarding the claim that Jin et al. (2024) addresses visual M-LLMs, when their work focuses on different architectures entirely. Additionally, while we appreciate Reviewer nLhj's perspective on the field's direction toward pure time-series foundation models, recent literature (including TimeGPT's mixed performance against traditional models) suggests this transition remains uncertain, making our methodological analysis particularly relevant. Reviewers CZBV and iE2G provided particularly valuable suggestions regarding experimental validation and hypothesis formulation. Reviewer hKqp raised important points about statistical testing and broader context.

We are particularly encouraged that multiple reviewers acknowledged our paper's strong experimental design and novel insights into multimodal LLMs for time series tasks. Their thorough feedback has helped us significantly strengthen both the theoretical foundations and empirical validation while maintaining our core contributions to examining LLM capabilities in time series analysis. We remain committed to incorporating these valuable insights while advancing the scientific understanding of AI systems' fundamental capabilities.

---

### Public Comment · ~Eamonn_Keogh1 · 2024-11-28
**Several of the reviewers suggested using "TSB-UAD", As [a] visually shows, you really cannot make any meaningful claims using that dataset.**

Several of the reviewers suggested using "TSB-UAD"
But it really is the case that "TSB-UAD" has unfixable problems: mislabeled data, triviality, repeated anomalies (leading to overcounting success), run-to-failure bias etc.
As [a] visually shows, you really cannot make any meaningful claims using that dataset.


[a] https://www.dropbox.com/scl/fi/cwduv5idkwx9ci328nfpy/Problems-with-Time-Series-Anomaly-Detection.pdf?rlkey=d9mnqw4tuayyjsplu0u1t7ugg&dl=0

---

### Meta-Review · Area_Chair_UJDx · 2024-12-20

**Metareview:**

This paper has been evaluated by 5 knowledgeable reviewers. Their opinions varied: 3 marginal acceptances, 1 marginal rejection and one straight rejection. The paper focuses on a timely topic and offers a broad set of hypotheses which are one by one more-or-less completely evaluated. The reviewers liked the interesting findings of value to research community. The authors provided extensive rebuttals, and some reviewers responded to them. One of the complaints by the reviewer who offered the lowest rating touched on the alternative between LLMs and dedicated time series foundation models. My personal observation is that the jury is still out with the question on which one should be exclusively used for this data modality. All things considered, I assess this paper as marginally acceptable for publication at ICLR.

**Additional Comments On Reviewer Discussion:**

There was no direct discussion among the reviewers, but it is clear that they have considered each other points reflected in their reviews. We had one comment from the public that challenged the utility of one particular benchmark dataset suggested by one of the reviewers for inclusion in the experiments.

---

### Decision · Program_Chairs · 2025-01-22

Accept (Poster)